# Inverse modeling of fire emissions constrained by smoke plume transport using HYSPLIT dispersion model and geostationary observations

Hyun Cheol Kim [1,2], Tianfeng Chai[1,2], Ariel Stein[1] and Shobha Kondragunta[3]

[1] Air Resources Laboratory, National Oceanic and Atmospheric Administration, College Park, MD, 20740, MD, USA
[2] Cooperative Institute for Satellite Earth System Studies, University of Maryland, College Park, MD, 20740, USA
[3] National Environmental Satellite, Data and Information Service, National Oceanic and Atmospheric Administration, College Park, MD 20740, USA

*Correspondence to*: Tianfeng Chai (tianfeng.chai@noaa.gov), Hyun Cheol Kim (hyun.kim@noaa.gov)

**Abstract.** Smoke forecasts have been challenged by high uncertainty in fire emission estimates. We develop an inverse
modeling system, the HYSPLIT-based Emissions Inverse Modeling System for wildfires (or HEIMS-fire), that estimates wildfire emissions from the transport and dispersion of smoke plumes as measured by satellite observations. A cost function quantifies the differences between model predictions and satellite measurements, weighted by their uncertainties. The system then minimizes this cost function by adjusting smoke sources until wildfire smoke emission estimates agree well with satellite observations. Based on HYSPLIT and Geostationary Operational Environmental Satellite Aerosol/Smoke Product
(GASP), the system resolves smoke source strength as a function of time and vertical level. Using a wildfire event that took place in the Southeastern United States during November 2016, we tested the system's performance and its sensitivity to varying configurations of modeling options, including vertical allocation of emissions and spatial and temporal coverage of constraining satellite observations. Compared with currently operational BlueSky emission predictions, emission estimates from this inverse modeling system outperform in both reanalysis (21 out of 21 days; -27% average root-mean-square-error
change) and hindcast modes (29 out of 38 days; -6% average root-mean-square-error   change) compared with satellite observed smoke mass loadings.

## 1   Introduction

Burning biomass is one of the major factors affecting global air quality (Crutzen and Andreae, 1990). Fire smoke plumes directly emit both particles that can impact cardiopulmonary health and precursors (e.g., $NO_x$, $SO_2$, $NH_3$, and volatile organic
carbons, or VOCs) (Andreae, 2019) that react to form secondary particulate matter (PM) or other pollutants, such as ozone (Dreessen et al., 2016; Jaffe and Wigder, 2012; Mok et al., 2016; Singh et al., 2012; Valerino et al., 2017). In addition to their impact on air quality, fire emissions influence direct and indirect radiative transfer, aerosol formation, and the formation of cloud condensation nuclei, and they further interact with clouds and, eventually, with the biosphere and climate. Interaction between fire and the climate is an important factor affecting the future direction of the environment (Bowman et
al., 2009). While high fire activities are affected by decadal-scale variation of the climate (Carvalho et al., 2011; Flannigan et

al., 2005; Spracklen et al., 2009), aerosols released from fires and changed surface albedo due to burnt areas are influential, as they disturb the radiative balance in the atmosphere (Liu et al., 2014).

Meeting National Ambient Air Quality Standards (NAAQS) requires U.S. state agencies to understand the primary emission sources for particulate matter. Notwithstanding many states' continuous efforts to control in-state sources of pollutants, it is
challenging to account accurately for fire emissions and the out-of-state transport of fire plumes. Due to the huge impact of fires on regional air quality, accurately forecasting their impact is an important public-service task, performed mostly by government agencies. The National Oceanic and Atmospheric Administration (NOAA) Smoke Forecasting System (SFS) was initiated after the large wildfire event in May 1998 (In et al., 2007; Rolph et al., 2009), to predict the movement of smoke from large wildfires (Rolph et al., 2009; Stein et al., 2009). The SFS uses the National Environmental Satellite, Data,
and Information Service (NESDIS) Hazard Mapping System (HMS) (Ruminski and Kondragunta, 2006) and the U.S. Forest Service's (USFS) BlueSky framework (Larkin et al., 2009) to detect fires and estimate emissions. The NOAA's HYSPLIT (Stein et al., 2015), a Lagrangian model which is designed to track air parcel trajectories, is then used to calculate transport, dispersion, and deposition of the emitted particulate matter. The SFS provides daily smoke forecasts over the continental United States, Alaska, and Hawaii to provide air-quality guidance to the public.

Eulerian systems are also used for smoke forecasting. Smoke emissions from wildfires have been incorporated into the NOAA's National Air Quality Forecast Capability system as real-time, intermittent sources; this system has been forecasting regional air quality for surface ozone and particulate matter concentration since 2015 (Lee et al., 2017). The High-Resolution Rapid Refresh Smoke (HRRR-smoke; https://rapidrefresh.noaa.gov/hrrr/) (Ahmadov et al., 2017) system also provides 36-hour forecasts for the continental United States using the WRF-Chem modeling system with emissions derived from
satellite-measured fire radiative power (FRP). Also, Chen et al. (2019) demonstrated an air quality forecast system over Canada that incorporates near-real-time measurement of biomass burning emissions to forecast smoke plumes from fire events. There are numerous other global smoke forecast systems (Chen et al., 2011; Larkin et al., 2009; Lee et al., 2019; Li et al., 2019b; Pavlovic et al., 2016; Sofiev et al., 2009).

Any improved smoke forecast system must confront several uncertainties, particularly fire (especially wildfire) emission
amounts and their allocation spatially and vertically. In general, fire emissions may be estimated in one of two ways: bottom up and top down. Regarding the more traditional bottom-up approaches, fuel consumption is estimated as the product of burnt size, pre-burn fuel loading of the fire-affected area, completeness of combustion, and emission factors (Seiler and Crutzen, 1980). Since emission factors are specific to the type of tree ablaze, a completely constructed database is very important in this approach. For example, Wiedinmyer et al. (2006), in estimating emissions from fires in North America,
estimated fuel loading based on a combination of satellite and ground-collected data, such as Moderate Resolution Imaging Spectroradiometer (MODIS) thermal anomalies, the Global Land Cover Characteristics dataset, the MODIS Vegetation Continuous Fields Product, and emission factors.

Recently, the global coverage of space-borne instruments has encouraged top-down approaches. A fire's heat signature (that is, FRP) is detectable by satellite, and FRP can be used to estimate the rate of combustion (Giglio et al., 2003; Kaufman et

al., 1998). Measurements of FRP from polar-orbiting sensors can both detect active fires and characterize their properties (Freeborn et al., 2009; Jordan et al., 2008; Schroeder et al., 2014). FRP data have been used to quantify biomass consumption, detect the locations of fire emission sources, trace gas and aerosol production (Ellicott et al., 2009; Kaiser et al., 2012; Vermote et al., 2009), and estimate the vertical extension of smoke plumes and other fire emissions (Val Martin et al., 2010). Several global fire emission databases (e.g., Global Fire Emissions Database (GFED), Fire Inventory form NCAR (FINN), Quick Fire Emissions Database (QFED), Global Fire Assimilation System (GFAS), Fire Energetics and Emissions Research (FEER), and Global Biomass Burning Emissions Product (GBBEP)) have been developed using one or both the bottom-up and top-down approaches (Ichoku and Ellison, 2014; Kaiser et al., 2012; van der Werf et al., 2010; Wiedinmyer et al., 2011; Zhang et al., 2012). Both bottom-up and top-down approaches have their own advantages and limitations. While the bottom-up approach may provide detailed information based on a process- or fuel-specific estimation, it relies on various surveys that require significant time and resources. On the other hand, the top-down approach relies on observations of a few atmospheric variables such as radiation or aerosol optical properties, but it has an advantage from its timely availability and geographical coverage. Both approaches complement each other for better fire emission estimation.

This study extends the current capabilities of the NOAA SFS fire smoke forecast systems, most of which estimate fire emissions using the surface and thermal characteristics of detected fire locations. Transport pathways of smoke plumes are rarely considered in determining emissions strength, vertical extension, and temporal variation. This study aims to develop an inverse modeling system for fire emissions based on a Lagrangian model that can resolve these transport pathways using HYSPLIT simulations and satellite observations. Such an approach has been adapted to estimate inversely various emission sources, including greenhouse gas emissions (Kunik et al., 2019; Nickless et al., 2018; Turnbull et al., 2019), volcanic ashes and sulfur dioxide emissions (Boichu et al., 2014; Crawford et al., 2016; Zidikheri and Lucas, 2020), and radionuclide release from nuclear power plant incident (Chai et al., 2015; Katata et al., 2015; Li et al., 2019a), but was rarely used in fire emission estimation (e.g. Nikonovas et al., 2017).

The remainder of the paper is structured as follows. Section 2 describes the model and satellite data used to detect fire locations and the transport of fire smoke. Section 3 concerns the methodology and structural design of the inverse modeling system. Results from a case study, sensitivity tests, and comparison with the currently operational system are presented in Section 4. Finally, Section 5 summarizes and discusses directions for future work.

## 2    Data

### 2.1    HMS and BlueSky

Consistent with the NOAA SFS system, the HMS data is utilized to detect wildfire information. HMS, developed as a tool to identify fires and their smoke emissions over North America in an operational environment, incorporates images from multiple geostationary and polar-orbiting environmental satellites, including Geostationary Operational Environmental Satellite (GOES)-East/West, Suomi-National Polar-orbiting Partnership (NPP), MODIS, and Advanced Very High

Resolution Radiometer (AVHRR) METOP-B, to provide the location and time of detected fires. Automated fire detection algorithms are first employed for each sensor, and then human analysts apply further quality control by examining visible channel imagery for false alarms and missed hotspots (Ruminski et al., 2008; Ruminski and Kondragunta, 2006; Schroeder et al., 2008).

The BlueSky system, developed by the U.S. Forest Service, provides the first guess for fire emission estimation. As a modeling framework, BlueSky links several models of fire information, fuel loading, fire consumption, fire emissions, and smoke dispersion (Larkin et al., 2009; Strand et al., 2012). For the original NOAA SFS system, BlueSky emissions are used as inputs for the dispersion model. In this study, we use the BlueSky emission rate as an initial guess before applying the inverse modeling system.

## 2.2 GASP

The GOES Aerosol/Smoke Product (GASP) is a retrieval of the aerosol optical depth (AOD) using GOES visible imagery (Kondragunta et al., 2008; Prados et al., 2007). This product is available at 30-minute intervals and 4 km × 4 km spatial resolution during the sunlit portion of the day. The Automated Smoke and Tracking Algorithm (ASDTA; https://www.ssd.noaa.gov/PS/FIRE/ASDTA/asdta_west.html) detects smoke associated with detected fire source locations. For each pixel, the radiative signatures of an aerosol layer (e.g. dust and smoke) are determined by the scattering and absorption properties of the aerosol. ASDTA also utilizes a pattern-recognition technique to isolate smoke aerosols from other type of aerosols, so it can recognize plumes transported far from fire sources. The GASP product is particularly useful for tracking fast-moving plumes, which polar-orbiting sensors often cannot detect since they provide only one daily image. Since the ASDTA product is a part of the GASP product, in this study, GASP and ASDTA indicate total AOD and smoke AOD, respectively. Hourly data were used for the inverse system.

## 2.3 HYSPLIT

HYSPLIT computes air parcel trajectories and the dispersion or deposition of atmospheric pollutants (Stein et al., 2015). It has been widely used to simulate pollutant events, including volcanic ash, smoke from wildfires, radioactive nuclei dispersion, and emissions of anthropogenic pollutants. For the inverse modeling system, we used the Transfer Coefficient Matrix (TCM) approach. The unit source calculations give the dispersion factors from the release point for every emission period to each downwind grid location, defining what fraction of emissions are transferred to each location varying as a function of time. This is defined as the TCM (Draxler and Rolph, 2012). The TCM is computed for inert and depositing species and, when quantitative air concentration results are required, the final air concentration is computed in a simple post-processing step that multiplies the TCM by the appropriate emission rate. Results for multiple emission scenarios are easily created and may be used to optimize model results as more measurement data become available.

For the inverse system, 120-hour HYSPLIT simulations were conducted daily, starting from 6Z using North American Model 12-km meteorology (NAM12), at each fire source location provided by the HMS fire detection information. Fifty

thousand particles were released for each simulation, and dispersed concentrations were vertically integrated up to 5000m onto 0.1 degree spatial grids. Hourly outputs were integrated to match with satellite observational data. HYSPLIT modelling options were configured to be consistent with the SFS system, including options for dry and wet depositions (i.e. 0.8 μm diameter with 2 $g/cm^3$ density) (Rolph et al., 2009). In this paper the integrated mass loading of particles from HYSPLIT simulation and satellite products will be compared with each other. For satellite products (e.g., GASP, ASDTA, and MODIS), smoke is converted from AOD using a simple conversion factor (i.e. 1 AOD = 0.25 $g/m^2$) which is compatible to 4 $m^2/g$ mass extinction efficiency (Nikonovas et al., 2017). Although we used a single conversion factor for the study, the actual conversion factors may vary in time and space (i.e. 3.9 – 5.3 $m^2/g$) (Chand et al., 2006; Hobbs et al., 1996; Ichoku and Ellison, 2014; Nikonovas et al., 2017; Reid et al., 2005). Therefore, applying more realistic conversion factors and their uncertainties into the system would be another factor in the future improvement of the system.

For HYSPLIT runs, smoke indicates the sum of dispersion simulations (i.e. TCM runs in concentration unit) multiplied by emissions for each source. Since we have integrated dispersion model outputs (in density unit) up to 5000m height, the results shown in the study are obtained by multiplying the column height (i.e. 5000m) and demonstrated as total mass loading for a column ($kg/m^2$).

## 3    Methodology

### 3.1    Overview

A HYSPLIT inverse system was built and successfully applied to estimate the cesium-137 releases from the Fukushima Daiichi Nuclear Power plant accident in 2011 (Chai et al., 2015). It was then modified to estimate the volcanic ash source strengths, vertical distribution, and temporal variations by assimilating MODIS satellite retrievals of volcanic ash clouds while using ash cloud top height information as well (Chai et al., 2017). It was found that simultaneously assimilating observations at different times produces better hindcasts than only assimilating the most recent observations. In this application, the HYSPLIT-based Emissions Inverse Modeling System for wildfires (HEIMS-fire) is designed to assimilate satellite observations to generate wildfire emission estimates. The smoke plume transport and dispersion, captured by frequent geostationary satellite retrievals, can be used as constraints to obtain the smoke emission estimates. In the system, a cost function quantifies the differences between HYSPLIT model predictions and satellite-observed AOD, weighted by model and observation uncertainties. Minimizing the cost function by adjusting emission rates at different fire locations and at several different release heights thereby provides the fire emission estimates.

### 3.2    Cost function

Taking a top-down approach, unknown emission terms are obtained by searching for the emissions that provide the model predictions that most closely match the observations. With fire locations mostly identified by the HMS system, unknown emission rates at these specified locations remain undetermined. At each fire location, released smoke can reach different

heights under various fuel-loading and meteorological conditions. In addition, emission rates may vary significantly with time. Thus, the unknown elements of the inverse problem are the emission rates $q_{ikt}$ at each wildfire location $i$ at different heights $k$ and time periods $t$. The cost function $F$ is defined as:

$$F = \frac{1}{2}\sum_{t=1}^{T}\sum_{k=1}^{K}\sum_{i=1}^{I}\frac{(q_{ikt} - q_{ikt}^{b})^2}{\sigma_{ikt}^2} + \frac{1}{2}\sum_{n=1}^{N}\sum_{m=1}^{M}\frac{(c_{nm}^{h} - c_{nm}^{o})^2}{\varepsilon_{nm}^2} + F_{other}$$

    where $c_{nm}^{o}$ is the $m$-th gridded satellite observation (e.g., GASP ASDTA smoke mass loading) at time period $n$, and $c_{nm}^{h}$ is its HYSPLIT counterpart.

    A background term is included to measure the deviation of the emission estimate from its first guess, $q_{ikt}^{b}$, obtained from the operational BlueSky emission computation. The background term ensures the problem remains well-posed even with the

limited observations available in certain circumstances. The background error variance $\sigma_{ikt}^2$ measures uncertainties in $q_{ikt}^{b}$. Pan et al. (2020) compared six global emission estimates and found that the total emission differs by a factor of 3.8. However, emission estimations at specific locations and times can have much larger errors. In addition, the vertical distribution of the smoke emissions is difficult to determine and this adds even more uncertainties to the emission estimates. We chose a large uncertainty for the background term as $\sigma_{ikt\_}$=1000×q $^{b}$ $_{ikt}$+ 1000 kg/hr at all locations and heights to

minimize the adverse impact of inaccurate BlueSky emission estimates. The observational error variances, $\varepsilon_{nm}^2$, represent uncertainties in both the model and observations, as well as the representative errors. Kondragunta et al. (2008) indicated that GOES aerosol retrievals over land were expected to have uncertainties within $0.15\tau \pm 0.05$, where $\tau$ is the AOD. Paciorek et al. (2008) showed a better performance of GOES aerosol retrievals in eastern U.S. than in western U.S. Green et al. (2009) demonstrated that GOES AOD correlates best with AERONET in autumn (September to November) than in other

seasons. They showed that the RMS error was 0.060 in autumn while the average for all seasons is 0.0149. Considering the better performance in the Eastern US and in November, AOD uncertainties of $0.10\tau \pm 0.06$ are assumed in this paper. A slightly larger additive component of the AOD error is chosen to include the effects of the representative errors and model errors which do not vary with the observed AOD values. $F_{other}$ refers to the other regularized terms that can be included in the cost function. For instance, Chai et al. (2015) has a temporal smoothness penalty term to avoid abrupt changes in the

temporal profile of the release rates. While this optimization problem could be solved to obtain optimal emission estimates using many minimization tools, we used the Limited-Memory Broyden–Fletcher–Goldfarb–Shanno (BFGS) algorithm (Zhu et al., 1997).

### 3.3 Inverse system

The HEIMS-fire system is designed to conduct a two-step operation: (1) estimation of fire emission using an inverse system, and (2) forecast modeling of fire smoke using estimated fire emissions. The inverse system utilizes observations and modeling systems available from multiple agencies. We aim to estimate objectively and optimally wildfire smoke sources' strength, vertical distribution, and temporal variations by assimilating GASP AODs. **Figure 1** summarizes the system's incremental stages of data processing, listed below with the required data (and their providing agencies):

(1) Fire detection: Hazard Mapping System (NESDIS)
(2) HYSPLIT simulations with unit emissions at different locations and release heights
(3) Construction of Transfer Coefficient Matrix with available observations
(4) Initial guess for fire emissions (BlueSky, U.S. Forest Service)
(5) Cost function minimization to estimate smoke emissions
(6) Smoke forecast using adjusted smoke emissions

By minimizing a cost function, the HEIMS-fire system provides adjusted fire emissions that can describe realistic smoke plumes. Results in the following section show that the assimilated smoke plumes agree well with satellite observations. The system requires as input a first guess before it can start to minimize the cost function. Selection of this input is usually critical both for the performance of the minimization calculation and for the final output.

We also explain here the naming conventions for temporal coverage of emission estimation processes and forecasting processes. This inverse system is designed to estimate fire emissions on the target day by analyzing past and present smoke field, and then utilizes them to forecast the future smoke field. The assimilation days (i.e. aday = 0,-1,-2) (see **Figure S1**) indicate the temporal coverage of dispersions and constraining observations. For a target day of November 13, inversions are conducted using HYSPLIT dispersion simulations and ASDTA observations for 24 hours (i.e. aday=0), 48 hours (i.e. aday=-1), 72 hours (i.e. aday=-2), and 96 hours (i.e. aday=-3). Estimated fire emissions are used to simulate fire smoke for November 13 (i.e. fday=0; reanalysis), and the same amount of fire emissions are used in forecast mode for November 14 (i.e. fday=+1) and 15 (i.e. fday=+2). Application of forecast mode will be further discussed in Section 4.5.

## 4 Results

### 4.1 Case study

A case study using a November 2016 wildfire event was conducted to test the performance of the HEIMS-fire system. This fire event was a series of wildfires in the southeastern United States in October and November 2016. The U.S. Forest Service reported at least 80,000 acres burned from October 23 to December 9, 2016. For the case study, we focused on the fire event that occurred in Georgia, South Carolina, North Carolina, and the adjacent states from November 10 to 17, 2016. **Figure 2** shows an example of fire smoke detected from MODIS true-color image and three AOD products from MODIS, GASP, and ASDTA on November 10, 2016. Wildfires in the Appalachians of northern Georgia, western North Carolina, and eastern

Tennessee began to produce large smoke plumes moving southeast. Numerous smoke plumes could be seen from active wildfires burning across the region. Changes in the fire events from November 8 to 19, 2016, are also shown as MODIS truecolor images in supplementary materials (**Figure S2**).

## 4.2   Model configuration

The month of November 2016 saw fires nationwide, although the most extensive fires happened in the southeastern U.S. region. We considered four geographic domains in determining fire source inputs, as shown in **Figure 3**. Red dots indicate HMS fire detections during November 2016. The results of the sensitivity test using these domains are discussed in Section 4.4.

The inverse modeling system was tuned using various sensitivity tests. A series of twin experiments was conducted to test the range of uncertainties that comes from the system design. A twin experiment is an idealized modeling test in which we assume that the modeled world adeptly mimics the real world. Using a true solution for the situation, we can test the system's capability to reproduce the true answer. We tested uncertainties of the system across multiple scenarios and four types of potential uncertainty (vertical allocation, temporal coverage, spatial coverage, and impact of observation errors) in these twin experiment cases. Detailed descriptions of the twin experiments and sensitivity test will be made available in a separate paper (See supplementary information).

## 4.3   Emission estimation

Fire emissions and their vertical distributions for each detected fire location were estimated using the HEIMS-fire system, inversely modeled from ASDTA AOD data as described above. Locations and times of fires detected by HMS were used to initiate HYSPLIT simulations, with emissions released over six layers (100, 500, 1000, 1500, 2000, and 5000 m). On November 11, 46 fire locations were identified within the assimilation domain (domain 1 in **Figure 3**). Thus, the TCM was established based on 276 HYSPLIT simulations (46 fires × 6 release altitudes) and GASP AOD observations. Emissions rates calculated from the BlueSky system were used as an initial condition. Emissions were evenly distributed to all layers used in the system.

Minimizing the cost function results in the estimation of fire emissions. **Figure 4** shows the agreement between the modeled and observed mass loading from the initial to the adjusted emission estimates. **Table 1** presents summary statistics for the changes in reconstructed smoke mass loading from the initial guess to the adjusted emissions. In the end, estimated fire emissions were combined to reconstruct fire smoke plumes. Using adjusted fire emissions, we can reconstruct the integrated smoke columns as a sum of adjusted emissions, $q_{ikt}$, applied to each TCM, $T_{ikt}$:

$$c(n,m) = \sum_{ikt} q_{ikt} \cdot T_{ikt}(n,m)$$

where $i$, $k$ and $t$ denote spatial, vertical and temporal allocation of emission sources, and $m$ and $n$ denote location and time of receptor (i.e. observations), respectively. **Figure 5** presents the spatial distribution of reconstructed fire smoke mass loading

for the case study in terms of column integrated density. We applied estimated fire emissions to TCM runs for each detected fire location and vertical release height and then merged them into one hourly concentration field. Reconstructed smoke
plumes (i.e., integrated dispersion outputs in 0-5000m height) show a good agreement with observed smoke (**Figure 5**). The first and second columns compare ASDTA and HEIMS smokes for spatially and temporally matching pixels, and the third column shows the full spatial coverage of HEIMS smoke for daytime (around 8AM-5PM local time) when ASDTA data is available. The November 17 output in **Figure 5** shows how the system responds when observations are limited or missing, although it still provides a robust result by honoring the initial guess information. On November 17, no ASDTA AOD was
provided from the satellite operation. Under 48-hour configuration (i.e. aday=-1), the inverse system still produced reasonable outputs using limited observations (November 16) and initial guess emissions (November 16 and 17). This case hints at the importance of both traditional (e.g. Blue Sky emissions) and new inverse system. They complement each other by one providing the latest data assimilation technique while the other providing a prior information and backup stability in a contaminated environment (e.g. excessive cloud cover).

## 4.4    Sensitivity tests

Similar to the twin experiments, which, as noted above, we will report in a forthcoming paper, we conducted a series of sensitivity tests to investigate how the inverse model responds to changes in input data and various configurations of the modeling framework. This will be achieved by focusing on variation in temporal coverage, spatial coverage, and vertical allocation of smoke plumes.
First, we changed the assimilation time windows from one (24-hours) to four days (96-hours). Since the impact of fire emissions easily translates over multiple days, we tested how temporal coverage affects system results. The 'one-day' (aday=0) simulation is run through the inverse model using dispersions and observations for the target day, while the 'two-day' simulation uses two days (i.e., 48 hours) of dispersions and observations (aday=-1). For this test, all observations within the assimilation time windows were selected for the assimilation and the evaluation. The results are shown in **Figure 6a**,
while the correlation and error statistics are summarized in the top section of **Table 2** (i.e. [A:24h, O:24h, E:24h], … , [A:96h, O:96h, E:96h]). With the exception of November 10 and 11, in the early stage of the fire event, both the correlation coefficient (R) and normalized root-mean-square error (NRMSE) were improved by the use of more days (i.e., three or four days) of dispersions and observations for the inverse model. This makes sense, because emissions from multi-day fire events spread out and affect concentrations over proceeding days.
A series of additional simulations were also conducted to test the system's sensitivity to the selection of observations for the assimilation and the evaluation. In these tests, we investigated combinations in assimilation time ("A" in **Table 2**), observational time ("O") and evaluation time windows ("E"). Results are also summarized in **Table 2**. For a fixed assimilation time period (i.e. [A:96h]), using shorter observational time window resulted in a better result. It is reasonable because we expect a better fitting with smaller number of data points. However, it can be easily exposed to overfitting
problem if available data for the assimilation is too small.

Second, we tested the layers at which fire emissions are initiated in the model. As expected, including more layers results in better statistics, since the transport and dispersion of each smoke plume can vary with the altitude to which their fire emissions are allocated. We tested the model's uncertainties on layers' maximum extension and resolution, with varying selections of two to seven layers at 100, 500, 1000, 1500, 2000, 5000, or 10000 meters. To test the maximum extension,

starting from two layers (i.e. with emissions released at 100 and 500 meters), we added the next higher layer over six test runs to investigate the effect of maximum extension of smoke plume. **Figure 6b** shows the results, and error statistics are summarized in **Table 3.** Including the 5000m layer, especially, resulted in noticeable changes, implying the potential benefit of including high-level transport for specific days. Since the 5000m layer is above typical planetary boundary layer height, emissions injected at this level experience different physical characteristics. Smoke lofted into the free troposphere is less

affected by turbulence and scavenging, and transports easily hundreds or thousands of kilometers downwind because of the higher wind speeds. Addition of the 5000m layer would better represent the potential long-range transport. Smoke plume rise is one of traditionally important questions in smoke modelling, so further research on the topic is warranted. Effects of the layer resolution were also tested. Starting from two layers (i.e. 100m and 5000m), we added intermediate layers up to six layers, and evaluated their performances (**Table S3**). As expected, including more layers resulted in the better statistics, but

its improvement was not significant after four layers.

In the third test, we varied the spatial coverage of input fire information. Although wildfire impacts easily spread by long-range transport, we could not include all the global fire information due to limited computational resources. We therefore tested different spatial domains of fire locations to evaluate what spatial coverage of wildfire detection information is required to estimate fire emissions. Fire sources inside domain 1 through 4 (**Figure 3**) were tested in the assimilation

constrained by ASDTA AOD inside Domain 1. **Figure 6c** and **Table 4** show correlation and error statistics from the sensitivity test of spatial coverage. In most days, we have better results when we include fire emission sources at least within domain 2. It makes sense considering the effects of transported fire plumes form Mississippi and Louisiana (**Figure 3**). Maximizing geographical coverage (e.g. domain 4) did not always result in the best performance in our case study. This result, however, should be taken carefully because we do not have strong fire activities outside domain 2 in our study case.

Strong long-range transport cases, typically form northwestern US, Canada and Alaska, would have bigger impacts.

### 4.5     Hindcast and operation

In this section, we conducted a HEIMS system for hindcast mode, and compared it with operational products from the SFS system. Both SFS and HEIMS use fire detection from HMS for consistency, and HEIMS uses SFS fire emissions for initial guess information. SFS simulates 72-hour dispersion of fire smoke for every day in November 2016, which is consistent

with fday=0,+1,+2 of the HEIMS hindcast simulations (as described in **Figure S1**).

Notable differences in the configuration of SFS and HEIMS are plume rise estimation, temporal resolution of fire emissions, fire decaying assumption, and meteorology. While SFS computes plume rise using the Briggs' equation (Arya, 1998; Briggs, 1969), which assumes an air parcel's rise is based only on the buoyance terms, HEIMS determines fire emissions' vertical

allocation using an inverse system. At the initial guess, SFS fire emissions are evenly distributed in all layers. Current
HEIMS assumes daily emission variation compared to hourly emissions of SFS. Also, SFS assumes 75% of emissions still happen at the same location the next day, the HEIMS uses 50% decay assumption after sensitivity tests, which will be discussed in the next section. For HEIMS simulation, we used aday=-1 (two-day temporal coverage) for the simulations shown. On the other hand, HEIMS would be benefitted with a better meteorology. Although both systems use the NAM12 forecast meteorology, HEIMS hindcast used the first 24 hours portion of everyday forecast cycle, and SFS used 72 hours
forecast.

For forecasting days, smoke is estimated as the summation of impact from previous days and new emissions on the target days. For example, smoke at fday=+2 can be reconstructed as

$$S_{fday=+2} = q_{f=0} \cdot TCM_{f=0} + q_{f=0} \cdot p \cdot TCM_{f=+1} + q_{f=0} \cdot p^2 \cdot TCM_{f=+2}$$


where $q$ and $p$ denote emissions and persistency rate, respectively. The persistency rate, $p$, assumes the change of future day emissions. Its role will be discussed in the next Section.

**Figure 7** and **Figure 8** demonstrate simulated fire smoke by SFS and HEIMS on Nov. 11 and two-day forecasts (hindcasts for HEIMS) for Nov. 12 and 13. Both systems reproduced well the smoke in their general patterns and intensity, as shown
in ASDTA AOD and MODIS truecolor image (**Figure 8).**

As expected, the HEIMS shows better agreement at fday=0 as fire emissions were assimilated on the day. For fday=+1 (i.e. Nov. 12) the HEIMS shows better agreement in RMSE and mean bias (RMSE=58.1 x10$^{-6}$ kg/m$^2$, bias = -22.4 x10$^{-6}$ kg/m$^2$ compared with RMSE=63.0 x10$^{-6}$ kg/m$^2$, bias = -42.2 x10$^{-6}$ kg/m$^2$) while SFS has better slope. For fday=+2, HEIMS is better in mean bias but worse in RMSE and R. **Table 5** summarizes RMSE statistics from HEIMS and SFS for each day of
November 2016. In most of days, HEIMS posts better statistics compared with SFS, implying the potential benefit of system improvement by adding an additional observational constraint. For the comparisons of HEIMS hindcast and SFS operational simulations, HEIMS system shows better performance in both hindcast days (16/19=84% on fday=+1 and 13/19=68% on fday=+2).

Change of fire activity is also a problem for both systems. If there is considerable change of fire activity for fday=+1 & +2,
the forecast will result in worse performance. If fire activities increase, the simulated smoke from HEIMS will be underestimated, and if fire activities decrease, the HEIMS system will overestimate the impact of smoke. Therefore, information of next day fire activity, or fire duration, will be important for an accurate fire smoke forecast system, which will be discussed further in the next section.

### 4.6    Persistency of fire activity

The selection of the persistent rate of daily fire emissions (i.e. persistency = 1-decaying_rate) and its importance to the smoke forecast system's performance are discussed here. In our current systems, both SFS and HEIMS, we use a simple

assumption of fire emission change for next day. **Figure 9** shows how the HEIMS responds to the selection of persistent rate for forecast days fday=+1 and +2. We applied five different persistent rates ranging from 0% to 100%; persistency=0% assumes no new fire occurs and persistency=100% assumes the same amount of fire emissions released at the same location from the previous day. For the top panel of **Figure 9,** simulated smokes in fday=+1 and +2 are sorely originated from fires in fday=0. On the other hand, persistency=100% simulation demonstrates accumulated impacts (target day and previous days), showing denser smokes estimated compared with persistency=0%.

An implication from these comparisons is the importance of persistent rate selection. Indeed, better smoke forecasting may require improvement via two separate steps. The first one is to estimate today's emissions, which can be improved by better assimilation techniques as we introduce in this paper. The second issue is to predict fire activity, which is more related to the studies of fire behavior. In more detail, we need to predict how long existing fires persist, and also to predict the occurrence of new fires, which may pose the greatest difficulty for daily operational systems. Without a better understanding and modeling of fire behavior, the current system has to rely on the empirical solution. For our case study, choosing persistent rate of 50%/day produced the best result, but it warrants further study with a long-term data set to be used in an operational system. Prediction of wildfire consistency based on the change of meteorological conditions, such as the Fire Weather Index (FWI, https://cwfis.cfs.nrcan.gc.ca/background/summary/fwi), will be a good indicator for the change of fire emission. Without this kind of fire behavior model, the fire smoke forecast system could be limited.

## 5    Summary and Discussion

Accurate estimation of emissions from wildfire sources is critical to improving the performance of air-quality forecast systems. Wildfire emissions may be estimated based on fire-detection information from the surface (bottom up) or instead based on the intensity of radiance measured from space (top down). This study extends the top-down approach by applying an additional constraint, i.e. transported smoke plume recorded by geostationary satellites. We developed an inverse modeling system to estimate wildfire smoke emissions over North America using NOAA's HYSPLIT and GOES Aerosol/Smoke products. This HEIMS-fire resolves the strength of smoke sources as a function of time and vertical level. The system adjusts estimated wildfire smoke emissions until they agree well with satellite observations.

We conducted numerous sensitivity tests, varying the temporal, vertical, and spatial coverage of the input data sets used to initiate the inverse system. Results are mostly consistent with general expectations based on the characteristics and behavior of fire events. As transport from previous days can impact large areas, including multiple days of observations to constrain fire emissions yields statistically better results. Including more vertical layers also leads to better results; for example, including the 5000-m layer especially resulted in the best improvement. Spatial coverage was tested in terms of four different domains, and while this particular test presented no solid conclusion, adding more information in general yielded better results, as expected. It also should be noted that the uncertainties of the emission estimation and the smoke forecasts

thereafter are not quantified in this study. An ensemble of HYSPLIT predictions using different meteorological inputs will be used to estimate the uncertainties of the results in the future.

For operational purposes, including an additional constraint that extends current smoke forecast systems to use smoke plume transport has clear advantages. Future study could improve this approach in several respects. First, the conversion of AOD to smoke mass loading is simply empirical; secondary formation of PM is not considered. Omission of chemical reaction models is the basic characteristic of trajectory- or dispersion-based models compared with Eulerian, full chemistry models. Applying estimated emissions to a chemistry dispersion model could improve results. Second, the system is highly

dependent on the quality of constraining observations. Use of the latest satellite instruments could further improve results. Third, we have not yet included surface observations into the inverse system. Utilizing both surface and more columnar observations from other satellite systems will improve the model performance. Fourth, we only used the target day fire emissions for smoke forecast. Since fire smokes last several days, including previous days' emissions will enhance the background effect.

This study aimed to improve the operational smoke forecast by providing accurate fire emission inputs. Unfortunately, the GASP product was discontinued in early 2018. However, the concept of minimizing a cost function based on satellite observations remains robust and can be applied to other data sets. In particular, we plan to apply the GOES-R Advanced Baseline Imager (ABI) product to constrain the extension of fire smoke.

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

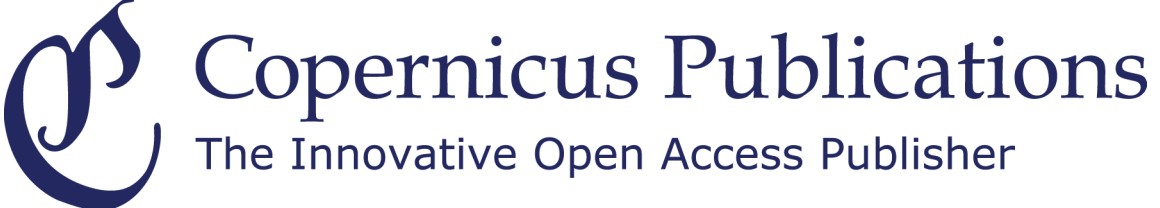

**Table 1. Performance evaluation. Statistics of modeled smoke mass loading (x10$^{-6}$ kg/m$^2$) using initial and top-down estimated fire emissions.**

|  |  | Nov. 10 | Nov. 11 | Nov. 12 | Nov. 13 | Nov. 14 | Nov. 15 | Nov. 16 | Nov. 17 |
|---|---|---|---|---|---|---|---|---|---|
| **mean** | Observation | 84.95 | 105.34 | 148.71 | 159.67 | 188.40 | 184.57 | 127.36 | 101.41 |
|  | Initial | 37.53 | 36.44 | 15.75 | 34.51 | 29.61 | 18.96 | 9.26 | 6.83 |
|  | Inverse | 73.72 | 77.92 | 108.20 | 105.24 | 122.15 | 114.85 | 87.76 | 66.14 |
| **RMSE** | Initial | 86.09 | 99.56 | 153.21 | 153.35 | 184.74 | 188.81 | 136.79 | 104.80 |
|  | Inverse | 54.07 | 69.37 | 94.40 | 104.85 | 124.88 | 130.89 | 92.35 | 70.77 |
| **NRMSE [%]** | Initial | 101.34 | 94.51 | 103.02 | 96.01 | 98.06 | 102.30 | 107.40 | 103.34 |
|  | Inverse | 63.64 | 65.85 | 63.48 | 65.67 | 66.28 | 70.92 | 72.51 | 69.79 |
| **Slope** | Initial | 0.88 | 0.74 | 0.33 | 0.54 | 0.46 | 0.24 | -0.15 | 0.26 |
|  | Inverse | 0.73 | 0.89 | 0.99 | 1.07 | 1.07 | 1.07 | 0.89 | 1.14 |
| **R** | Initial | 0.4 | 0.46 | 0.02 | 0.18 | 0.28 | 0.25 | -0.08 | 0.00 |
|  | Inverse | 0.65 | 0.62 | 0.31 | 0.48 | 0.44 | 0.35 | 0.15 | 0.15 |


**Table 2. Sensitivity test for temporal coverage. Uses of assimilation (A), observation (O), and evaluation (E) time windows (24 hours to 96 hours) were tested, and statistics, NRMSE and R, were compared. The best performance is marked as bold.**

| | NRMSE [%] | | | | R | | | |
|---|---|---|---|---|---|---|---|---|
| Coverage | A: 24h O: 24h E: 24h | A: 48h O: 48h E: 48h | A: 72h O: 72h E: 72h | A: 96h O: 96h E: 96h | A: 24h O: 24h E: 24h | A: 48h O: 48h E: 48h | A: 72h O: 72h E: 72h | A: 96h O: 96h E: 96h |
| Nov. 10 | 66.6 | 63.7 | **62.7** | 68.7 | **0.677** | 0.654 | 0.669 | 0.577 |
| 11 | 75.2 | 65.9 | 64.0 | **63.7** | 0.410 | **0.620** | 0.587 | 0.593 |
| 12 | 75.1 | 63.5 | 61.4 | **60.6** | 0.373 | 0.309 | **0.562** | 0.526 |
| 13 | 72.8 | 65.7 | 59.9 | **59.6** | 0.298 | 0.476 | 0.397 | **0.576** |
| 14 | 75.6 | 66.3 | 61.0 | **56.8** | 0.285 | 0.444 | **0.522** | 0.469 |
| 15 | 82.7 | 70.9 | 62.7 | **60.2** | 0.105 | 0.348 | 0.465 | **0.511** |
| 16 | 69.8 | 72.5 | 68.2 | **61.7** | 0.146 | 0.150 | 0.372 | **0.478** |
| 17 | | 69.8 | 72.5 | **68.2** | | 0.146 | 0.150 | **0.372** |
| Coverage | A: 24h O: 24h E: 24h | A: 48h O: 48h E: 24h | A: 72h O: 72h E: 24h | A: 96h O: 96h E: 24h | A: 24h O: 24h E: 24h | A: 48h O: 48h E: 24h | A: 72h O: 72h E: 24h | A: 96h O: 96h E: 24h |
| Nov. 10 | 66.6 | 49.1 | 49.0 | **38.7** | 0.677 | 0.713 | 0.716 | **0.716** |
| 11 | 75.2 | 80.6 | 78.4 | **78.4** | 0.410 | 0.621 | 0.623 | **0.623** |
| 12 | 75.1 | **45.2** | 56.5 | 55.8 | 0.373 | 0.406 | 0.425 | **0.430** |
| 13 | 72.8 | 58.0 | **52.9** | 63.5 | 0.298 | 0.534 | 0.581 | **0.589** |
| 14 | 75.6 | 54.6 | 53.8 | **53.1** | 0.285 | 0.567 | 0.593 | **0.609** |
| 15 | 82.7 | 43.9 | **42.4** | 44.6 | 0.105 | 0.554 | **0.589** | 0.571 |
| 16 | 69.8 | 34.3 | 28.4 | **27.4** | 0.146 | 0.464 | **0.507** | 0.505 |
| 17 | | | | | | | | |
| Coverage | A: 96h O: 24h E: 24h | A: 96h O: 48h E: 48h | A: 96h O: 72h E: 72h | A: 96h O: 96h E: 96h | A: 96h O: 24h E: 24h | A: 96h O: 48h E: 48h | A: 96h O: 72h E: 72h | A: 96h O: 96h E: 96h |
| Nov. 10 | **53.0** | 62.1 | 62.1 | 68.7 | **0.746** | 0.657 | 0.657 | 0.577 |
| 11 | **54.7** | 61.9 | 63.7 | 63.7 | 0.583 | **0.615** | 0.593 | 0.593 |
| 12 | **43.0** | 54.0 | 59.1 | 60.6 | 0.425 | 0.481 | **0.545** | 0.526 |
| 13 | **40.3** | 46.4 | 54.5 | 59.6 | **0.664** | 0.510 | 0.500 | 0.576 |
| 14 | **38.3** | 43.5 | 50.5 | 56.8 | **0.688** | 0.603 | 0.499 | 0.469 |
| 15 | **43.9** | 50.8 | 55.3 | 60.2 | **0.647** | 0.572 | 0.533 | 0.511 |
| 16 | **33.5** | 42.2 | 53.2 | 61.7 | 0.667 | **0.685** | 0.581 | 0.478 |
| 17 | | 36.1 | 53.8 | 68.2 | | **0.624** | 0.520 | 0.372 |
| Coverage | A: 96h O: 24h E: 24h | A: 96h O: 48h E: 24h | A: 96h O: 72h E: 24h | A: 96h O: 96h E: 24h | A: 96h O: 24h E: 24h | A: 96h O: 48h E: 24h | A: 96h O: 72h E: 24h | A: 96h O: 96h E: 24h |
| Nov. 10 | 53.6 | 47.1 | 47.1 | **38.7** | **0.746** | 0.715 | 0.715 | 0.716 |
| 11 | **58.1** | 81.0 | 78.4 | 78.4 | **0.668** | 0.624 | 0.623 | 0.623 |
| 12 | **41.3** | 48.0 | 56.7 | 55.8 | **0.489** | 0.432 | 0.433 | 0.430 |
| 13 | **39.5** | 49.8 | 55.4 | 63.5 | **0.691** | 0.612 | 0.588 | 0.589 |
| 14 | **39.5** | 45.8 | 51.9 | 53.1 | **0.698** | 0.612 | 0.612 | 0.609 |
| 15 | **37.8** | 43.1 | 43.1 | 44.6 | **0.682** | 0.552 | 0.566 | 0.571 |
| 16 | **33.7** | 29.6 | 27.9 | 27.4 | **0.614** | 0.542 | 0.510 | 0.505 |
| 17 | | | | | | | | |


**Table 3. Sensitivity tests for selection of two to seven layers among 100, 500, 1000, 1500, 2000, 5000, and 10000 meters. Beginning from two layers at 100 and 500 meters, the next higher layer is added for each test. The best performance is marked as bold.**

|  | Date | 2 layers (100,500m) | 3 layers (100-1000m) | 4 layers (100-1500m) | 5 layers (100-2000m) | 6 layers (100-5000m) | 7 layers (100-10000m) |
|---|---|---|---|---|---|---|---|
| NRMSE [%] | Nov. 10 | 74.1 | 69.9 | 65.7 | 64.5 | **63.7** | 63.7 |
|  | 11 | 82.2 | 75.6 | 70.1 | 68.4 | **65.9** | 65.9 |
|  | 12 | 68.6 | 65.6 | 67.0 | 68.9 | **63.5** | 63.5 |
|  | 13 | 76.0 | 73.2 | 72.8 | 72.8 | 65.7 | **65.3** |
|  | 14 | 78.1 | 77.1 | 76.3 | 75.5 | **66.3** | 66.3 |
|  | 15 | 81.1 | 75.4 | 73.2 | 71.4 | **70.9** | 70.9 |
|  | 16 | 84.6 | 80.8 | 76.4 | 73.7 | 72.5 | **72.5** |
|  | 17 | 81.3 | 78.3 | 73.5 | 73.2 | **69.8** | 69.8 |
| R | Nov. 10 | 0.58 | 0.62 | 0.64 | 0.65 | **0.65** | **0.65** |
|  | 11 | 0.49 | 0.55 | 0.60 | 0.61 | 0.62 | **0.62** |
|  | 12 | 0.23 | 0.21 | 0.15 | 0.12 | **0.31** | 0.31 |
|  | 13 | 0.43 | 0.44 | 0.43 | 0.43 | **0.48** | 0.47 |
|  | 14 | 0.43 | 0.41 | 0.41 | 0.43 | **0.44** | **0.44** |
|  | 15 | 0.30 | 0.34 | 0.34 | **0.35** | 0.35 | 0.35 |
|  | 16 | 0.11 | 0.10 | 0.12 | 0.15 | 0.15 | **0.15** |
|  | 17 | **0.40** | 0.34 | 0.33 | 0.24 | 0.15 | 0.15 |

**Table 4. Sensitivity tests for spatial coverage. Use of observational data in terms of spatial availability is tested. Domains 1-4 are shown in Figure 3. The best performance is marked as bold.**

|  | Date | Domain 1 | Domain 2 | Domain 3 | Domain 4 |
|---|---|---|---|---|---|
| NRMSE [%] | Nov. 10 | 64.5 | 65.9 | 65.7 | **63.7** |
|  | 11 | 64.5 | **62.3** | 65.3 | 65.9 |
|  | 12 | 59.7 | **59.5** | 59.8 | 63.5 |
|  | 13 | 60.1 | 60.1 | **60.1** | 65.7 |
|  | 14 | 62.3 | 63.6 | **62.3** | 66.3 |
|  | 15 | 66.5 | **66.4** | 66.5 | 70.9 |
|  | 16 | **66.5** | 72.7 | 72.6 | 72.5 |
|  | 17 | **66.5** | 72.7 | 72.6 | 69.8 |
| R | Nov. 10 | 0.48 | 0.62 | 0.62 | **0.65** |
|  | 11 | 0.48 | **0.63** | 0.62 | 0.62 |
|  | 12 | **0.32** | 0.32 | 0.31 | 0.31 |
|  | 13 | 0.47 | 0.47 | 0.47 | **0.48** |
|  | 14 | 0.43 | 0.42 | 0.43 | **0.44** |
|  | 15 | 0.33 | 0.33 | 0.32 | **0.35** |
|  | 16 | **0.33** | 0.15 | 0.15 | 0.15 |
|  | 17 | **0.33** | 0.15 | 0.15 | 0.15 |

Table 5. Performance statistics, RMSEs, for HEIMS and SFS smoke mass loading for 0-2 forecast days (e.g., fday=0,+1,+2). For each forecast day, the better performance from two systems is marked as bold. Statistics for days without observations or without operational outputs are available. [Unit: $10^{-6}$ kg/m$^2$]

| Date | fday = 0 | | fday = +1 | | fday = +2 | |
|---|---|---|---|---|---|---|
| | HEIMS | SFS | HEIMS | SFS | HEIMS | SFS |
| November 1 | **64.5** | 105.8 | **54.6** | 70.5 | **40.2** | 77.0 |
| 2 | **53.6** | 70.6 | **52.0** | 76.2 | **51.8** | 53.9 |
| 3 | **61.4** | 76.4 | **52.8** | 54.5 | **74.3** | 74.4 |
| 4 | **45.1** | 53.3 | **67.2** | 71.6 | 106.8 | **86.5** |
| 5 | **41.6** | 70.4 | **64.3** | 82.4 | **68.5** | 72.2 |
| 6 | **54.2** | 86.1 | **61.6** | 95.7 | - | - |
| 7 | **56.3** | 88.3 | - | - | 75.2 | **72.9** |
| 8 | - | - | 74.8 | - | 45.0 | - |
| 9 | **60.4** | 73.3 | **39.7** | 40.3 | **69.3** | 70.5 |
| 10 | **22.8** | 57.0 | 61.2 | **60.1** | 71.8 | 89.1 |
| 11 | **45.5** | 60.3 | **58.1** | 63.0 | 95.4 | **71.0** |
| 12 | **39.9** | 60.7 | 103.0 | **69.1** | 79.4 | 88.1 |
| 13 | **52.7** | 74.3 | **69.7** | 83.8 | 75.3 | **61.4** |
| 14 | **60.3** | 81.9 | **61.8** | 63.9 | **42.0** | 44.2 |
| 15 | 57.8 | - | 31.4 | - | - | - |
| 16 | **26.6** | 39.8 | - | - | - | - |
| 17 | - | - | - | - | **46.2** | 47.0 |
| 18 | - | - | 52.3 | **45.6** | 45.0 | **43.6** |
| 19 | **35.6** | 49.7 | **44.4** | 45.8 | **42.3** | 46.0 |
| 20 | 42.8 | - | 41.3 | - | 77.3 | - |
| 21 | **35.0** | 45.2 | **64.5** | 77.0 | **63.7** | 76.6 |
| 22 | 49.2 | - | 54.3 | - | 43.2 | - |
| 23 | **60.7** | 75.5 | **42.7** | 46.2 | 92.7 | **92.4** |
| 24 | **39.8** | 50.9 | **89.8** | 93.3 | **70.4** | 71.1 |
| 25 | **87.6** | 93.4 | **71.7** | 72.1 | 102.1 | **101.6** |
| 26 | **62.3** | 72.0 | **90.6** | 100.8 | - | - |
| 27 | **73.1** | 99.8 | - | - | - | - |
| 28 | - | - | - | - | - | - |
| 29 | - | - | - | - | - | - |
| 30 | - | - | - | - | **40.0** | 40.0 |
| HEIMS Performance | 21/21=100% | | 16/19=84% | | 13/19=68% | |

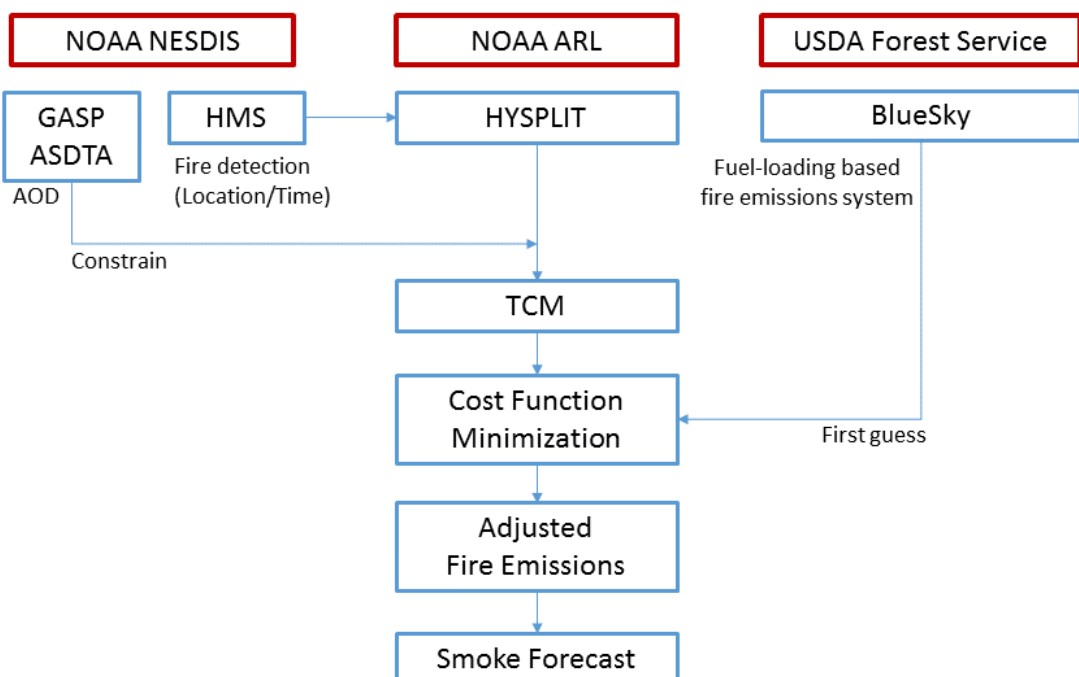

**Figure 1. Schematic diagram of the HYSPLIT-based Fire Emission Inverse Modeling System.**

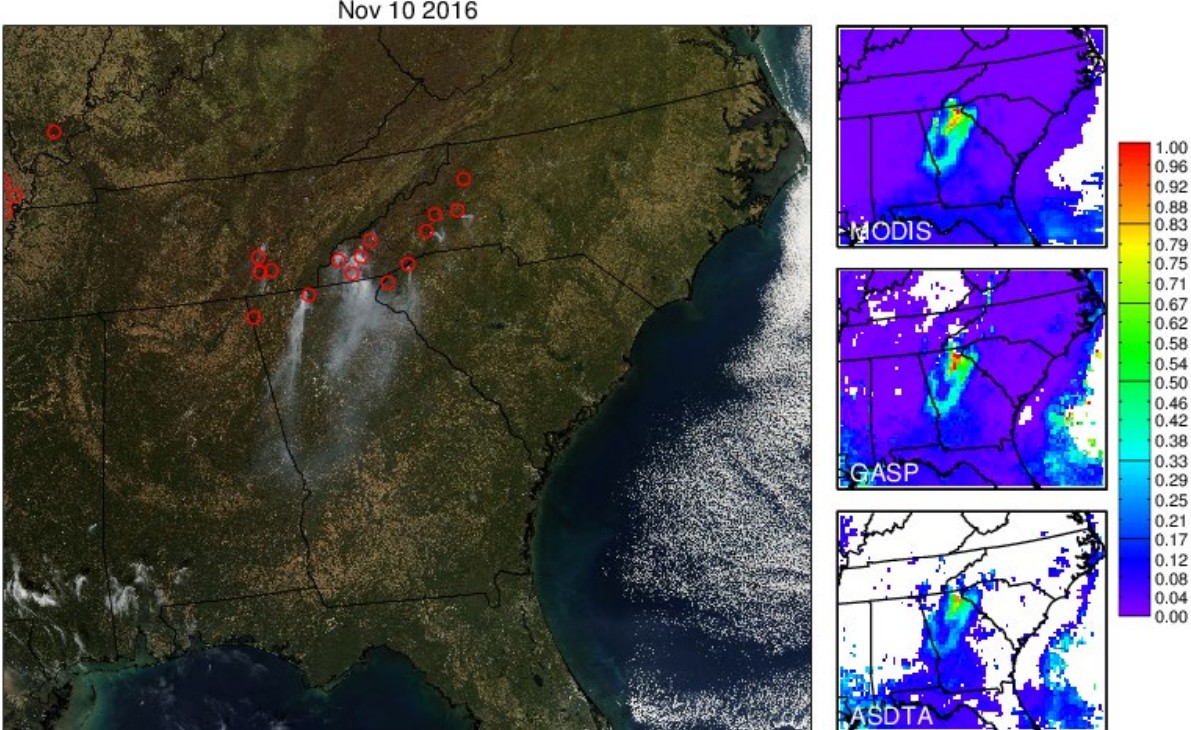

**Figure 2. Detection of fires over the southeastern region of the United States on November 10, 2016. Truecolor image from MODIS (left), MODIS AOD (top right), GASP AOD (middle right), and ASDTA AOD (bottom right) are shown. MODIS truecolor images and AOD are obtained from earthdata.nasa.gov, and GASP and ASDTA AOD are obtained from NOAA NESDIS.**

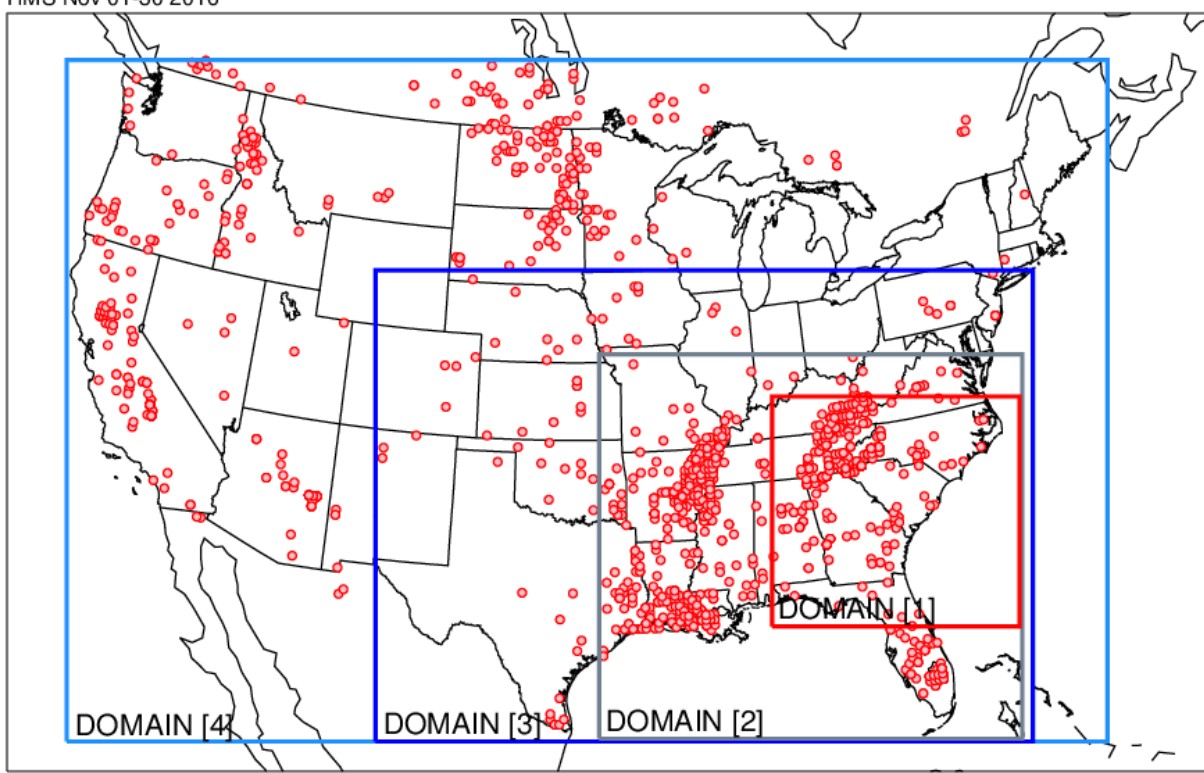


**Figure 3. Geographical coverage of case study domains. Red dots indicate HMS fire detections during November 2016. The four labeled domains indicate spatial coverage of fire source inputs for the inverse system used in the sensitivity tests.**

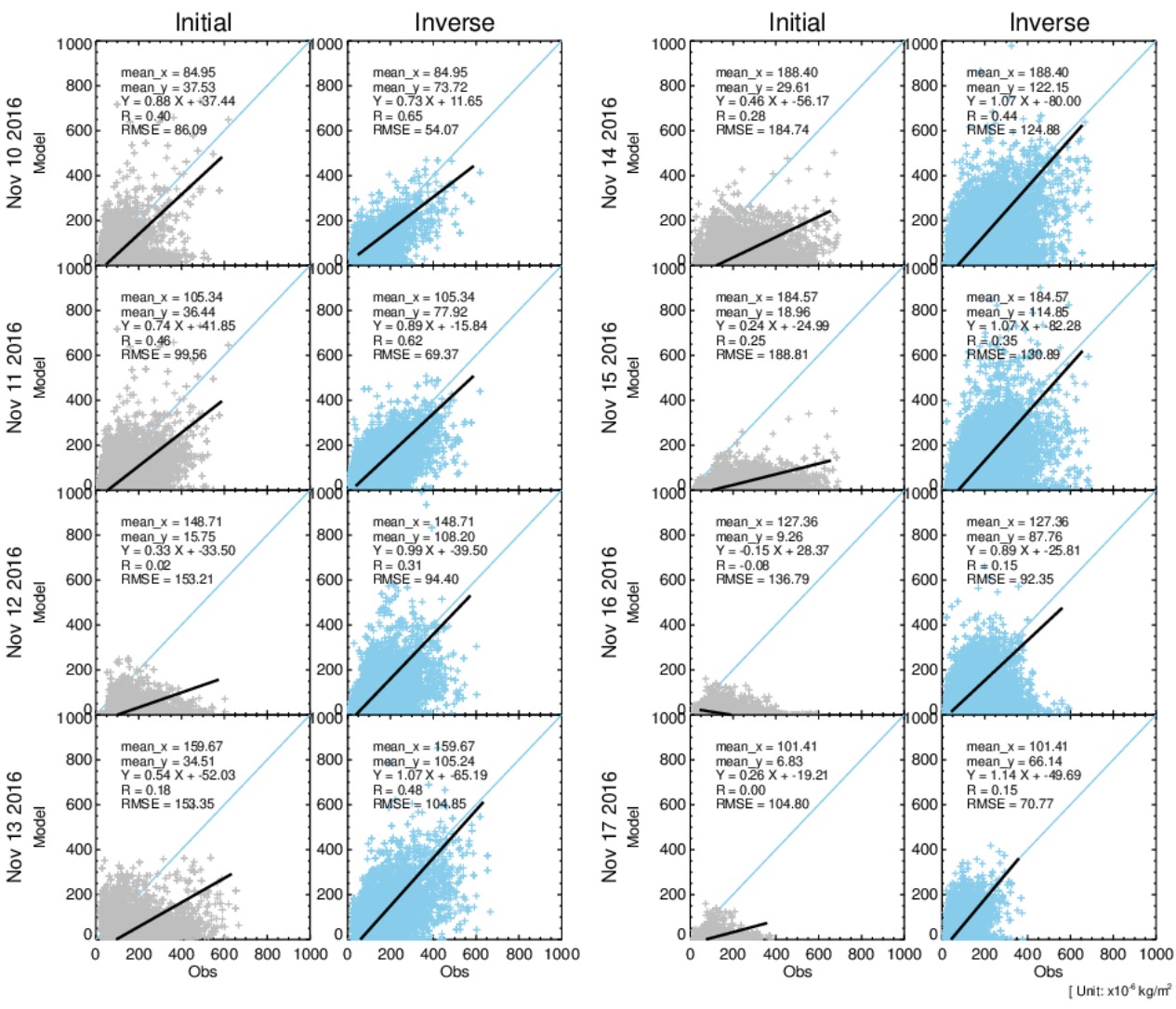

**Figure 4.** Scatter plot comparison between initial and assimilated smoke mass loading using adjusted fire emissions. A 48-hour observation (aday=-1) and 6-layer plume release configuration is used. [Unit: $\times 10^{-6}$ kg/m$^2$]

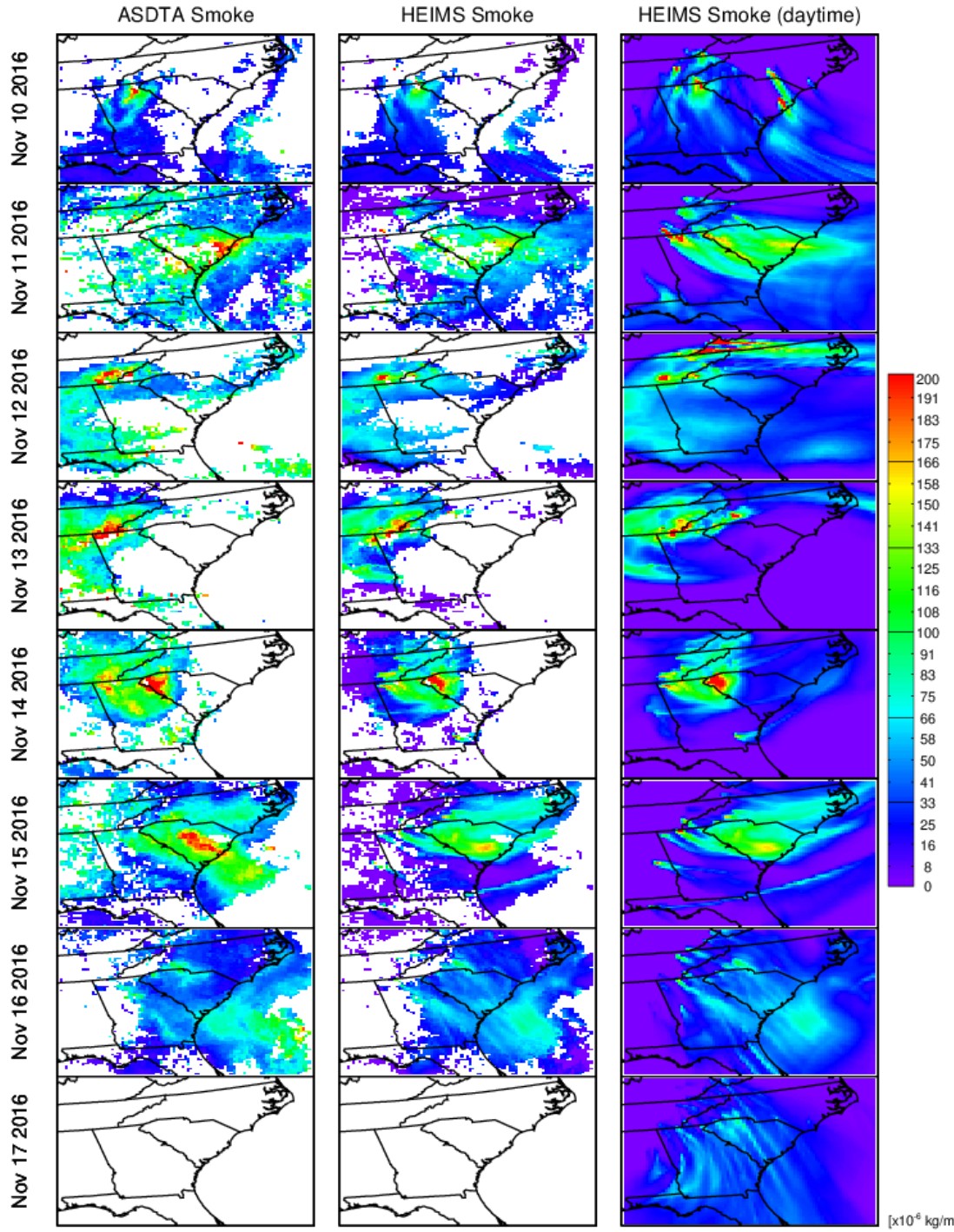

Figure 5. Comparison of observed and reconstructed smoke plumes during November 10-17, 2016. Smoke mass loading from ASDTA (left) and reconstructed HEIMS smokes for ASDTA-matching (middle) and for daytime (right). A 48-hour observation (aday=-1) and 6-layer plume release configuration is used.

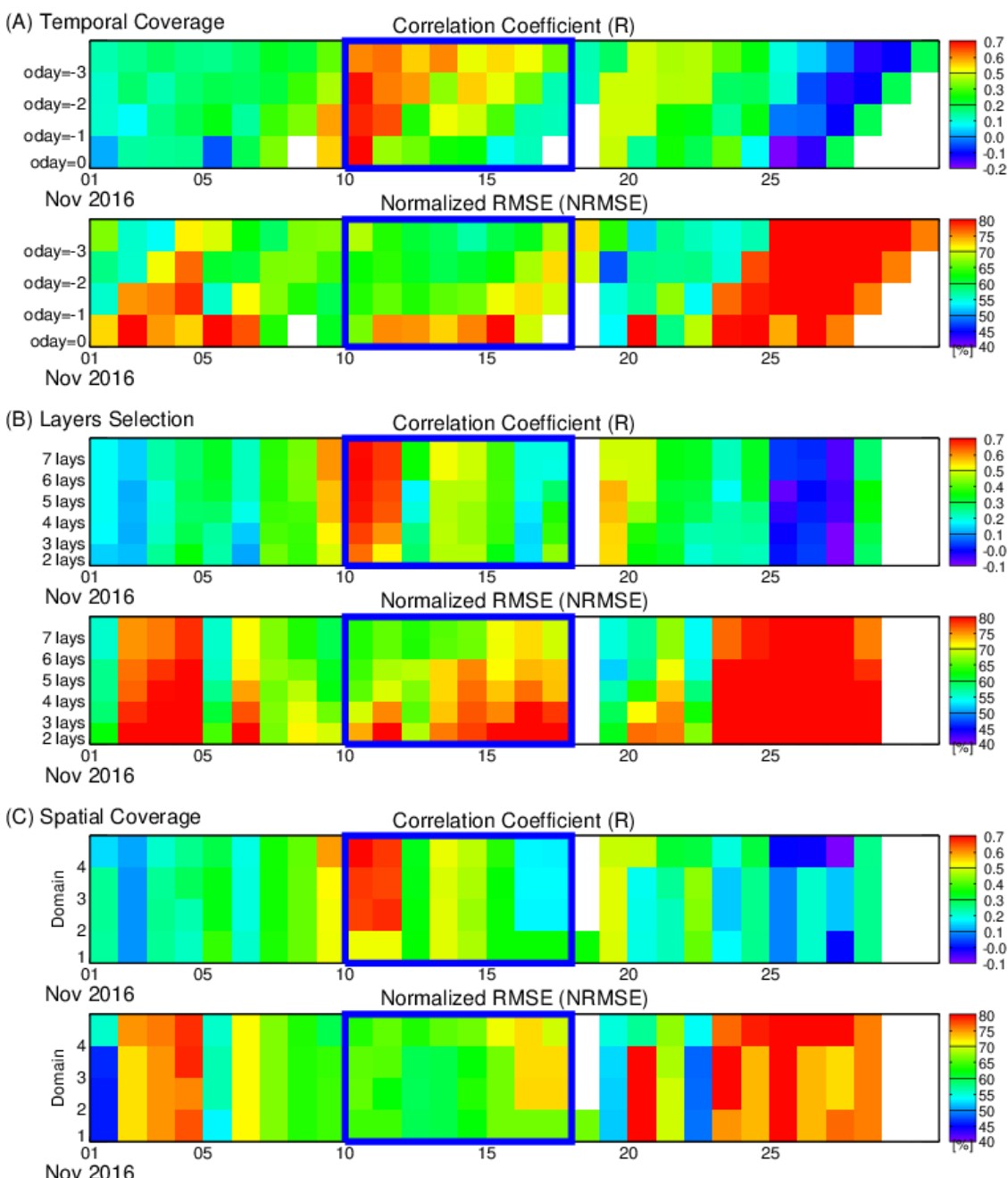

Figure 6. Sensitivity tests for (a) temporal coverage, (b) layer selection, and (c) spatial coverage. Temporal coverage of inputs is tested for between 24 hours (aday=0) and 96 hours (aday=-3). Selection of layers is tested using two to seven layers among 100, 500, 1000, 1500, 2000, 5000, and 10000 meters. Beginning from two layers at 100 and 500 meters, the next higher layer is added for each test. Spatial coverage is tested through domain 1 to 4. The thick blue boxes indicate the intensive fire episode period, November 10-17, 2016.


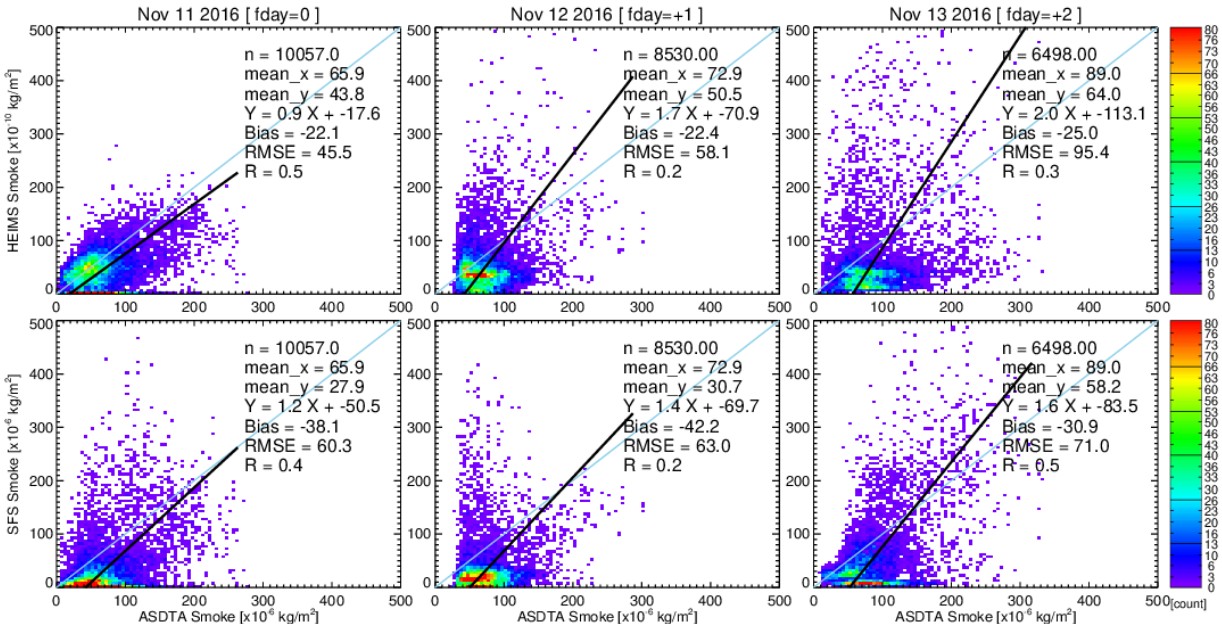

**Figure 7. Scatter plot comparisons between ASDTA smoke and HEIMS smoke (upper) and ASDTA and SFS smoke (lower) for forecast days, fday=0, +1, +2.**


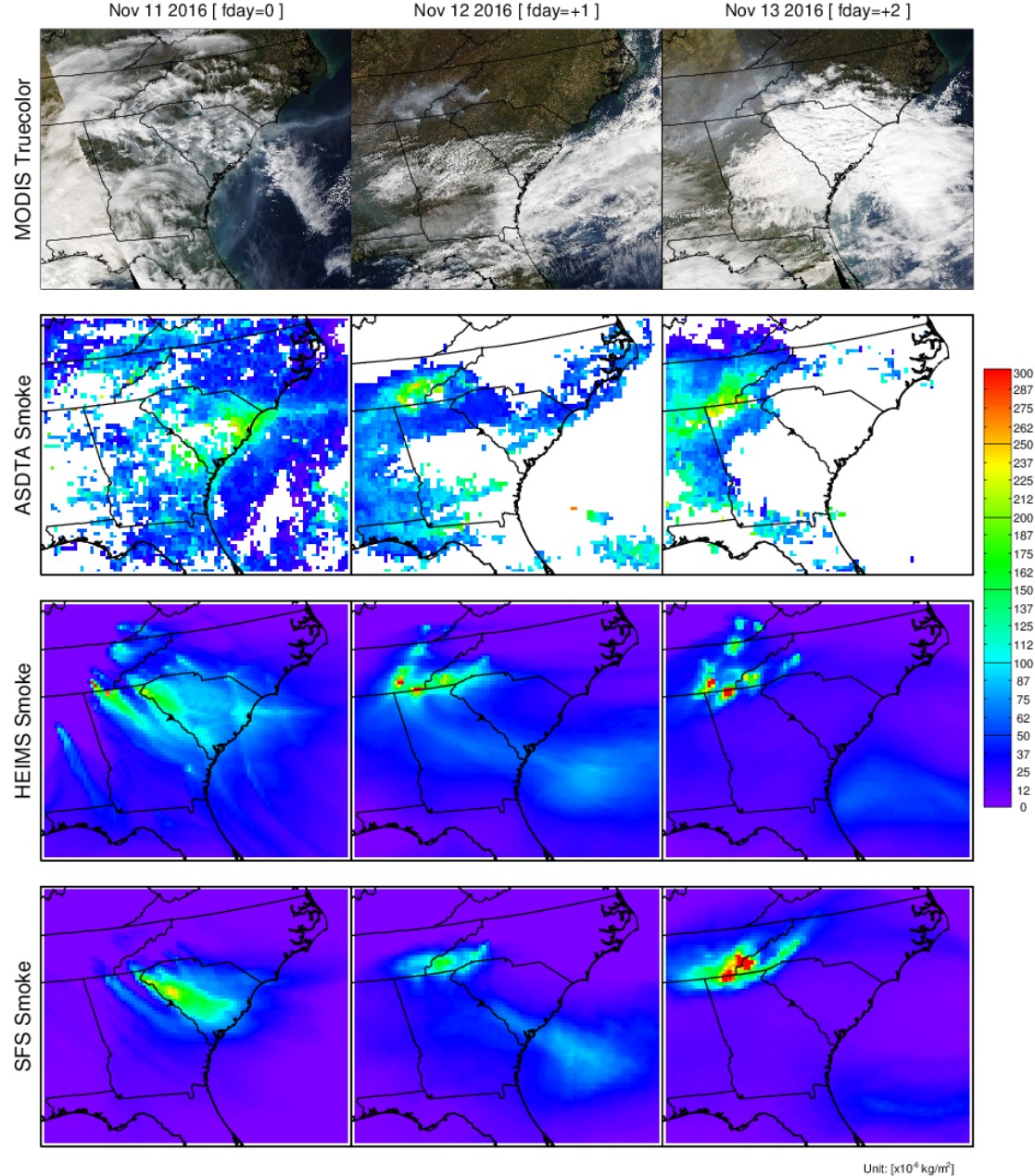

**Figure 8. Spatial distributions of observed and forecasted fire smoke plumes on November 11-13, 2016; Truecolor image from MODIS (1st row), ASDTA smoke (2nd row, converted from AOD), HEIMS smoke hindcast (3rd row), and SFS smoke forecast (4th row, from operation) are shown.**

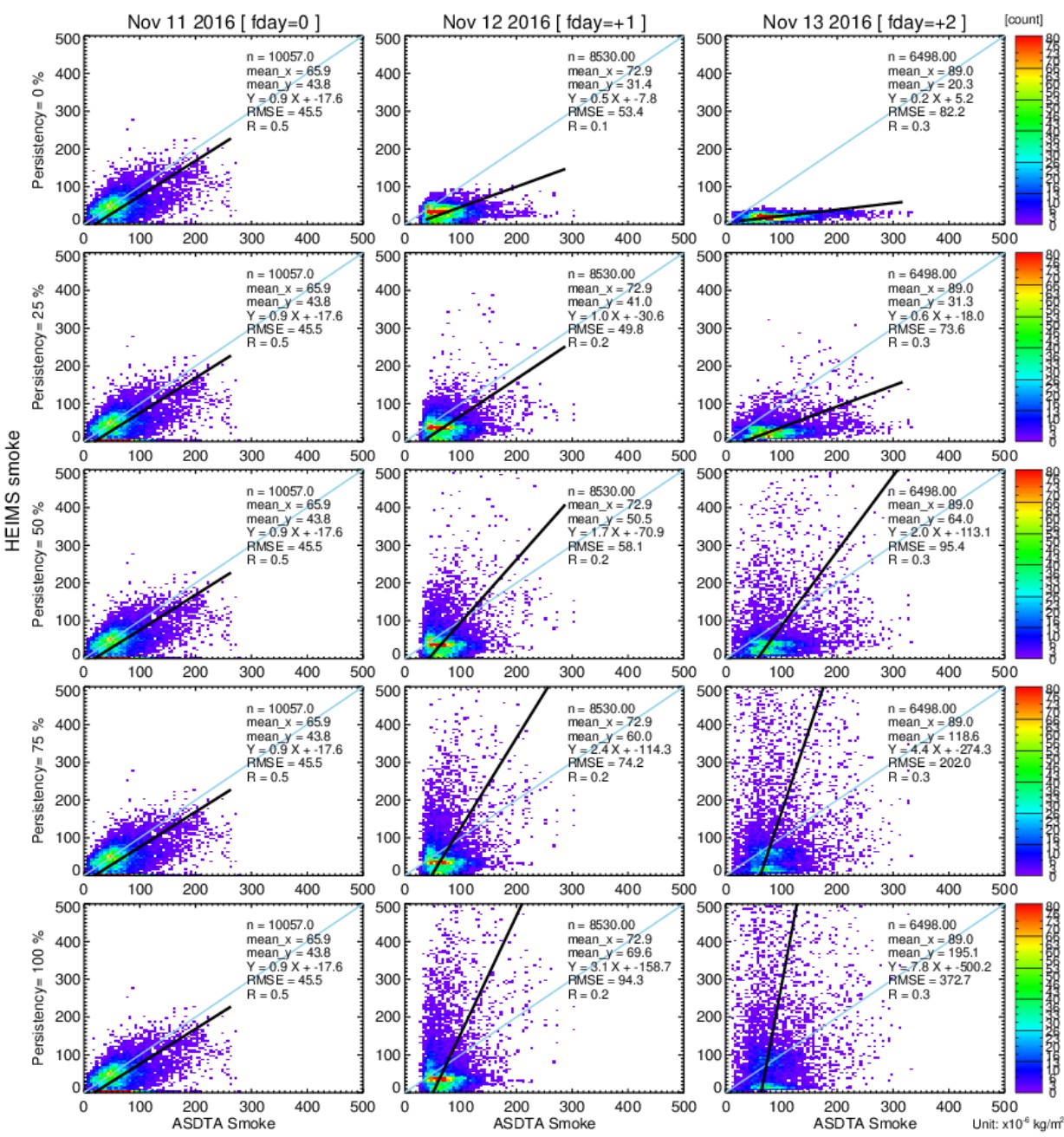

**Figure 9. Sensitivity test with varying persistent rates. For p =75%, we assume 75% of fday=0 emissions last in fday=+1, and 75%^2 emissions last in fday=+2.**