# Peer review of "Inverse modeling of fire emissions constrained by smoke plume transport using HYSPLIT dispersion model and geostationary observations"

_Atmospheric Chemistry and Physics, 2020_

## Referee Comment (RC1) · Anonymous Referee #1 · 15 Apr 2020

**General comments**

The manuscript by Kim et al. entitled "Inverse modeling of fire emissions constrained by smoke plume transport using HYSPLIT dispersion model and geostationary observations" investigated how to improve fire emission quantifications by assimilating transported smokes with Lagrangian model simulations. The results provide a pathway to improve operational model forecast of fire smoke, which is of critical importance to accurate air-quality assessment. The study described in this paper is attractive and the case experiments were executed well. However, there are still some issues to be addressed and questions to be clarified, which have been listed as follows.

**Specific comments**

- In section 1 (Introduction), there is no background information of applications of inverse modeling on fire emissions estimations. As this is the focus of this work, it would be helpful to inform readers about previous investigations on this topic, as well as to highlight the features and advantages of the inverse modeling system proposed in this work.
- The HYSPLIT model used to compute dispersion factors using the TCM approach is described in section 2.3, but some details are not provided, e.g., the temporal resolution of the HYSPLIT integration and the TCM results. Also, deposition considered by using a radioactive decay constant. Is it equivalent to the deposition process of fire smoke aerosols? How does it compare with the deposition considered in other Eulerian air quality models with more comprehensive chemistry?
- In the HYSPLIT simulations, dispersed concentrations were vertically integrated up to 5000 m to get partial column mass loading of smoke particles. Is there any reference for the selection of the column height (5000 m)? In addition, smoke loading from satellite observations is converted from AOD using a constant conversion factor. Could the authors further discuss the possible uncertainties that could be introduced by using this constant conversion factor? For example, how is this constant conversion factor compare with values reported by previous literature, and is there any relation between the conversion factor and other plume features (e.g., plume age)? And, this uncertainty can be considered in observation errors in the inverse modeling system.
- In section 3.2 of the cost function used in the inverse system, the estimations of error have not been provided, which are important terms for the inversion method. Firstly, how are the background error variances evaluated, and how does the value used here compare with typical uncertainties of fire emissions? Secondly, as mentioned in section 2, the observation error variances are composed of uncertainties in Lagrangian model, observations, as well as representative errors. It would be important to include more details about the determination of these error components. Thirdly, what kind of error terms should be considered in $F_{other}$ and how are they determined?

- The ASDTA smoke AOD data are assimilated in the HEIMS-fire system to obtain inverse estimation of fire emissions. As indicated in the introduction, the ASDTA data are based on automatic detections of fire smoke plume and represent smoke AOD, which means that the background AOD has been subtracted from total value. Is it correct? Then could the authors explain how is the background AOD derived? Also, the uncertainties in background AODs can impact inversion results, because an overestimation of background AOD will lead to underestimated smoke AOD (then underestimated fire emissions) and vice versa. This uncertainty can also be considered in the observation errors used for inversion.
- In section 3.3, the naming conventions of inversion and forecasting processes are described. It's a bit inaccurate to say that "fire emissions on November 13 can be estimated using ASDTA observations for 24 hours (i.e. oday=0), 48 hours (i.e. oday=-1), 72 hours (i.e. oday=-2), and 96 hours (i.e. oday=-3)." Technically, if we focus on fires on the target day (oday = 0), the regional impact and transported smoke plumes from those fires would be found on the following days (oday=1, 2, 3, etc.). Therefore, in this case the emissions on November 13 can only be constrained by the observations on oday=0 in reanalysis mode. On the other hand, the observations on oday=0 can constrain emissions on oday=0, as well as emissions on previous days simultaneously. So, the major benefit of including more observational days is getting more constraints for fires on multiple days, and providing a better estimation of the background smoke plume for the target day. In this case, it would be more precise to say that "for a target day of November 13, inversions are conducted using ASDTA observations for 24 hours (i.e. oday=0), 48 hours (i.e. oday=-1), 72 hours (i.e. oday=-2), and 96 hours (i.e. oday=-3)."
- Column particle mass loading is used to constrain fire smoke emissions, and emissions released at different numbers of vertical layers are tested in the sensitivity analysis. Including 5000 m level make an obvious improvement for the results. But there is a lack of analysis for the reason. A possible reason is that, 5000 m is usually above the Planetary Boundary Layer Height (PBLH). So, it would be interesting to examine the PBLH and plume injection height for this case, since smoke injection height is important to smoke transport. Smoke lofted into the free troposphere is often transported hundreds or thousands of kilometers downwind because of the higher wind speeds, generally lower turbulence levels, and less scavenging processes at higher altitudes. While, smoke trapped within the PBL is usually well mixed, and remains near the source region. If most of the fire spots in this case showed injected emissions above the PBLH, then it means that, including 5000 m in the simulations allows a better representation of emission injection, and the plume can be transported further and better constrained by observations.
- As a follow-up of the last comment, could it provide better results by tuning the emission release heights incorporating information from a plume rise model?
- For the sensitivity test on the time range of observation data used in inversion, "The 'one-day' (oday=0) simulation is run through the inverse model using dispersion and

observations for the target day, while the 'two-day' simulation uses two days (i.e., 48 hours) of dispersion and observations (oday=-1)". As this sensitivity test focus on the time range of observations, I think it is unfair to compare the results using different days for both of dispersion and observations. For example, if we compare the results using 'one day' and 'two days' shown in the current test, for the 'one-day' simulation, all the mismatches of modeled and observed smoke mass loading would be attributed to adjustments of fires on the target day, which would likely lead to a significant error in the emission estimates for the target day. While the 'two-day' simulation allows the fires on oday=-1 to be constrained at the same time. Therefore, it would make more sense to use four days of dispersion for all the simulations for this test, and just change the observation days from 1 (24 hours) to 4 (96 hours).

- At the end of section 4.3, it is mentioned that "The November 17 output shows how the system responds when observations are limited or missing, although it still provides a robust result by honoring the initial guess information". But no result is referred. It would be clearer to add a reference to the figure/table supporting this sentence. If it is Figure 4, it would also be better to indicate the number of points in each panel to show that the observations are limited on Nov 17, given that many points could be overlapped.

- P8, L249-255, this paragraph can be more concise. Also, a description of the result of the spatial coverage sensitivity test is missing.

- As stated in section 4.5, for the forecasting days, smoke is estimated as the summation of impact from fires existed on previous days and new emissions on the target days, and fire plumes initialized on fday =0, 1, 2 are used here for the summation. How do the authors determine how many previous days should be considered? Since the impact of fires can extend to multiple days, would it give a better result by adding the contribution of smokes initiated on the analysis days, e.g. oday = -1, -2, and -3?

**Technical corrections**

- In the abstract, it is concluded that the inverse modeling system developed here outperform than the current operational forecast product in terms of RMSE, but it's not clear RMSE of which variable is denoted here, and what observation dataset are the hindcast results and operational product evaluated against.

- P3, L92: "A modeling framework" -> "As a modeling framework"

- P4, L95: emission rate or emission? Do they represent the same term in this paper (i.e. fire smoke particles emission)? It seems that both are used throughout this paper. It would be better to use one of them and keep consistent.

- P4, L125: I do not quite understand why "the results shown in the study are obtained by multiplying the column height (i.e. 5000 m)" here. It has been shown that, the column TCM is calculated by integrating dispersed concentrations vertically, so the TCM is already in units of column loading per unit emission. Is it correct?

- P7, L189: true color image. There is not a true color channel.
- P7, L214: "estimation of assimilated fire emissions" -> "estimation of fire emissions"
- Section 4.3, the time range of observations assimilated in this case experiment is not indicated.
- What's the date for the result shown in Figure 5?
- P8, L225: remove "that"
- P8, L242: "smoke dispersions" -> "fire emissions"
- P8, L247, "As expected, including more layers generally produce better result." This sentence is nearly a duplication of the sentence in L242-243.
- P10, L295: "From the top panel of Figure 9" -> "For the top panels of Figure 9"
- P10, L295: "are solely originated fires" -> "are solely originated from fires"
- P10, L303: "reply on" -> "rely on"
- P10, L312, for the "additional constraint", "transported smoke plume" could be better. There are other places of this term, please consider revising them accordingly.

---

## Referee Comment (RC2) · Meelis Zidikheri (Referee) · 17 May 2020

General overview: This paper uses an inverse modelling approach to estimate fire emissions at various locations as a function of time and height by making use of satellite measurements of smoke optical depth. The method was applied to a November 2016 wildfire event in the USA. Various sensitivity tests were performed including the effect of different satellite observation time windows, maximum source height, and domain size. The results were compared to results obtained with the smoke forecasting system currently in use and it was found to yield better agreement with observations

as expected. Having estimated the fire emissions, forecasts (or rather hindcasts) were produced by making use of these optimal emissions. It was found that the forecasts were generally better than corresponding forecasts obtained by the conventional system, but the improvements were modest in my view, particularly for longer lead times. Nevertheless, I think this is a solid piece of work, and would recommend that it be accepted for publication. However, there are some issues that need to be addressed by the authors as listed below, which in my view would improve the paper. In addition, in my view the manuscript would benefit from another round of proofreading. Main issues: 1. How the experiments with different assimilation time windows are run needs to be made clearer. Is it the case that in all experiments the emissions are initiated at the same locations and times, the only difference being the satellite data assimilation time window? If that is the case, can you clarify why longer time windows results in better agreement with observations for the analysis? I would have thought it would be easier to fit the model to data from one timestep than to fit the model to data from different timesteps. I can not quite understand the explanation given in the paper (e.g. Line 240), which I interpret to mean that this is due to emissions at (x1,t1) affecting the smoke field at (x2,t2). This is true, but I would expect better results for the smoke field at (x2,t2) if there was no observation at (x1,t1) because the emission at (x1,t1) would only be constrained by the observation at (x2,t2) and not (x1,t1) and (x2,t2) simultaneously. You probably also need to clarify which data points are included in the verification statistics (for example, when you assimilate data on day 0, do you calculate verification statistics based on day 0 only or do you also include days -1, 2 etc?). 2. In performing the forecasts, it is not completely clear whether you are using NWP analysis data to drive the forecasts, or you are using NWP forecast data. If this system is to be used operationally, then clearly NWP forecasts will need to be used, which we expect will result in poorer forecasts than when the NWP analyses are used. I think this needs to be clarified. 3. Nothing has been said about analysis and forecast errors. Can this system be used to predict errors (uncertainty) or is it purely deterministic? I think some comments on this issue in the paper would be useful. Detailed comments: 1. In the abstract you have several acronyms that are only defined later in the text. These must be defined in the abstract or alternatively think about removing them altogether from the abstract. In my view they are not that important for the abstract and can be removed. For example, is it important to know that the model has been developed by NOAA in the abstract? 2. Line 32: Remove capital letters from words 2-6 in the sentence starting with "Meeting...". 3. Line 43 (or thereabouts): You need to state that HYSPLIT is a Langrangian model (and maybe elaborate on what means) as in the next paragraph you start talking about Eularian models. 4. I also notice that you don't leave spaces between paragraphs, which makes it hard to read at times. Fix this if you can. 5. Line 54: You distinguish "top-down" and "bottom-up" approaches to estimating fire emissions. I think it would useful as well to discuss the advantages and disadvantages of these approaches somewhere in the introduction. 6. When discussing the methodology it might be worthwhile to explicitly mention that the "inverse system" enables the calculation of fire emissions that produces an "analysis" smoke field covering the observational time window (near past to present) and that these optimal emissions are then used to forecast the future smoke field. I do not think this has been made explicit although it may be obvious to the expert reader. 7. Another point that I think needs to be made clear in the methodology section is how the experiments with different observational time windows (oday = 0, -1, 2 etc) differ from each other in the model setup. For example, for oday = 0, is it that there are no emissions prior to day 0, or is it that emissions prior to day 0 are unconstrained (i.e. they use prior values), or is it that the emissions prior to day 0 are in the model but they are constrained by observations on day 0 only. I suspect it is the last scenario, but this needs to be made clear. You could include this information in the paragraph starting at Line 177, which I think needs more clarity. For example, you could indicate that oday = 0 means that emissions from the model start time until day 0 (for example, November 13) are constrained by observations on day 0 only; oday = -1 means that emissions from the model start time until day 0 are constrained by observations on day 0 and day -1 (November 12) etc. As I mentioned, it is not clear that is indeed the experimental setup, but whatever it is, it needs to made clear. When

you calculate the verification statistics for the oday = 0 type experiment, for example, do you include all data (for days 0, -1, -2 etc) or just data for day 0? Please make this clearer. 8. It would also be useful in the context of Section 3 (or elsewhere like Figure 1 or in the conclusion) to say something about how the two systems (Blue Sky/Initial? and Inverse) really need to complement each other in practice. Apart from providing the all-important first guess (or 'prior') to the inverse system, I would imagine that the Blue Sky system would also be needed to provide the first operational forecast when there are not yet any (or sufficient) observations to assimilate. In addition, in situations where there is extensive cloud cover, I would imagine that there would not be observations of smoke that could be used. This comment is related to Comment #5 above. 9. Line 219: The notation associated with this equation needs to be improved. Although I think I understand what you are trying to say, as it stands it is ambiguous. Firstly, it is not clear what "smoke" is supposed to be. I think you are looking at smoke mass loading, or it could be optical depth. Whatever it is, it needs to be made clear. I would use a single-letter symbol to represent the smoke variable. Since "c" is used in the first equation (Lines 148-149) that is what should be used here as well for consistency. Similarly, I would use a single letter to represent the TCM matrix coefficient (T seems like a good choice). Apart from that the equation doesn't look mathematically correct. Summing over the indices i, k, t should produce a scalar, independent of i, k, t but we know that the smoke field (mass load or optical depth) is a function of location, i, and time, t. So something is missing. I think some more work needs to be done here so that the equation makes sense. 10. Line 226: change "during" to "when". 11. Line 226: Last sentence of paragraph. Some more needs to be said here. Why is there no data on November 17? In addition, you are implying that no satellite data was used in the analysis (presumably because there was none – you must clearly state that if that is the case, and if so, why). And yet in the caption to Figure 5 we are told that a 48-hour observation time window was used. If that is the case, why could not the data for November 16 be used to constrain the analysis? Is it because your algorithm does not perform optimisation if any time steps are missing in the observational time

window? You might have mentioned this elsewhere, but how frequent are the satellite observations? Every hour? Every 10 minutes? 12. Paragraph starting at Line 234: This is related to Comment #7. If you are using emissions initiated at the same time in all experiments in which you vary the length of observational time window, it is surprising, I think, that you get a better fit with a longer observational time window. Intuitively, all things being equal, you would expect to get a better fit with a smaller number of data points during the analysis. Of course, for the forecast you would expect better results with more data points during the analysis because you are less exposed to overfitting problems with more data points. It looks like you're talking about the analysis here so can you make it clearer why using data from earlier times (say day -1, -2 etc) makes the analysis better on day 0 for example? I find that a bit unclear. I can see that if you are using data from all days to calculate your forecast verification statistics and not only on the days that data is assimilated, you would get better results by using more data because you are better constraining the smoke over a longer period. If that is the case, please make it clearer which data points are included in the verification statistics. 13. Paragraph starting at Line 242: When I first read this, it sounded like you are varying the vertical resolution of the sources in these experiments. But is turns out you are just varying the maximum heights of the sources. I think you should consider rewording this so that it is immediately clear what you are doing. In some sense, I think varying the vertical resolution and keeping the maximum height the same would be a more interesting experiment. Rising plumes tend to have a neutral buoyancy level at which most of the mass tends to be detrained from the plume. This is true for volcanic plumes; I don't know if it's true here. So, changing the vertical resolution may have a significant impact on the results. 14. Line 255: You must say something about the outcome of the "third test" here. Would you say coverage does not matter? Or is it better for bigger coverage (Domains 3 and 4)? If the results are inconclusive, say so. 15. Line 271: You need to say something about "p" here even if briefly; for example, "p" is the persistence rate that measures . . . as explained further in Section xxx. 16. Lines 298-307. Given the significant uncertainty in "p", an ensemble modelling approach might be fruitful in

the forecast mode. Have you looked at this? If not, it might be something you could look at in the future. 17. Line 325: What is "PM"?

---

## Author Comment (AC1) · 28 Jun 2020

**"Inverse modeling of fire emissions constrained by smoke plume transport using HYSPLIT dispersion model and geostationary observations"**

**General response**

Authors express their appreciation to the two reviewers and the editor. Thanks to their productive comments, we were able to improve our manuscript. As the reviewer recommended, we have included additional tests and discussions on the sensitivity. Mistakes were revised, and performance statistics were also updated. We provide below the general responses and the point-by-point responses to the reviewer's comments. Reviewers' comments are shown in italics.

Here are three major points in the revised version of the manuscript.

1. Sensitivity tests for temporal coverage

As both reviewers commented, we agree that the temporal sensitivity section needs a clarification of the goal. Our original intention was to find the most efficient temporal coverage of the inverse system to conduct the assimilation process, which includes both time windows for the assimilation and the use of observational data. On the other hand, the reviewers asked to test the sensitivity by the change of observational data use. We believe that both tests are meaningful, so we included both test results in the manuscript. The former is important in terms of the efficiency of the inverse system because dispersion simulations to construct the TCM is the most time-consuming processes in the inverse system. The latter, selection of observations (used in the assimilation), is also important to investigate the responses of the inverse system.

Indeed, there are three adjustable time windows in the processes and evaluation of fire emission assimilation. As shown in Figure R1, the assimilation (or analysis) time window denotes the temporal coverage of fire emission sources and simulated dispersions. The observational data time window (or selection of observations used for constraining in the assimilation) can be set equal to or shorter than the assimilation time window. The evaluation time window (or selection of observations used for evaluation) is also set equal to or shorter than the observational time window. Therefore, we tested combinations of these time windows.

- 1) Assimilation time window (source, dispersion and assimilation process)
- 2) Observation time window (observation)
- 3) Evaluation time window (model and observation)

Model performance statistics were calculated with combinations of temporal coverages of assimilation (shown as "A" in Table R1; 24h-96h), observation (shown as "O", 24h-96h), and evaluation (shown as "E", 24h-96h). Results are summarized in Table R1. As discussed in the manuscript, including more analytical period generally improves the performance of the inverse system, but the difference is not critical for more than 48-hour time window.

With fixed analytical period (A:96h), performance statistics are better with shorter observational time window applied. It makes sense because we expect to have a better fit with smaller number of data points during the analysis process although it also has a risk of potential overfitting problem, as the reviewer #2 commented.

The manuscript was revised to include these additional sensitivity tests.

(Line 270) "First, we changed the assimilation time windows from one (24-hours) to four days (96-hours). Since the impact of fire emissions easily translates over multiple days, we tested how temporal coverage affects system results. The 'one-day' (aday=0) simulation is run through the inverse model using dispersions and observations for the target day, while the 'two-day' simulation uses two days (i.e., 48 hours) of dispersions and observations (aday=-1). For this test, all observations within the

assimilation time windows were selected for the assimilation and the evaluation. The results are shown in **Figure 6a**, while the correlation and error statistics are summarized in the top section of **Table 2** (i.e. [A:24h, O:24h, E:24h], ..., [A:96h, O:96h, E:96h]). With the exception of November 10 and 11, in the early stage of the fire event, both the correlation coefficient (R) and normalized root-mean-square error (NRMSE) were improved by the use of more days (i.e., three or four days) of dispersions and observations for the inverse model. This makes sense, because emissions from multi-day fire events spread out and affect concentrations over proceeding days.

A series of additional simulations were also conducted to test the system's sensitivity to the selection of observations for the assimilation and the evaluation. In these tests, we investigated combinations in assimilation time ("A" in **Table 2**), observational time ("O") and evaluation time windows ("E"). Results are also summarized in **Table 2**. For a fixed assimilation time period (i.e. [A:96h]), using shorter observational time window resulted in a better result. It is reasonable because we expect a better fitting with smaller number of data points. However, it can be easily exposed to overfitting problem if available data for the assimilation is too small."

Figure R1 Two steps for smoke forecast with the HEIMS-fire system. Temporal coverage of assimilation days (aday=0,-1, - 2,-3) and forecast days (fday=0,+1,+2) for operational tests. Assimilation process includes three time windows – assimilation, observation and evaluation time windows.

|          |        | NRMS   | SE (%) |           |        | I      | ર      |        |
|----------|--------|--------|--------|-----------|--------|--------|--------|--------|
|          | A: 24h | A: 48h | A: 72h | A: 96h    | A: 24h | A: 48h | A: 72h | A: 96h |
| Coverage | O: 24h | O: 48h | O: 72h | O: 96h    | O: 24h | O: 48h | O: 72h | O: 96h |
|          | E: 24h | E: 48h | E: 72h | E: 96h    | E: 24h | E: 48h | E: 72h | E: 96h |
| Nov. 10  | 66.6   | 63.7   | 62.7   | 68.7      | 0.677  | 0.654  | 0.669  | 0.577  |
| 11       | 75.2   | 65.9   | 64.0   | 63.7      | 0.410  | 0.620  | 0.587  | 0.593  |
| 12       | 75.1   | 63.5   | 61.4   | 60.6      | 0.373  | 0.309  | 0.562  | 0.526  |
| 13       | 72.8   | 65.7   | 59.9   | 59.6      | 0.298  | 0.476  | 0.397  | 0.576  |
| 14       | 75.6   | 66.3   | 61.0   | 56.8      | 0.285  | 0.444  | 0.522  | 0.469  |
| 15       | 82.7   | 70.9   | 62.7   | 60.2      | 0.105  | 0.348  | 0.465  | 0.511  |
| 16       | 69.8   | 72.5   | 68.2   | 61.7      | 0.146  | 0.150  | 0.372  | 0.478  |
| 17       |        | 69.8   | 72.5   | 68.2      |        | 0.146  | 0.150  | 0.372  |
|          | A: 24h | A: 48h | A: 72h | A: 96h    | A: 24h | A: 48h | A: 72h | A: 96h |
| Coverage | O: 24h | O: 48h | O: 72h | O: 96h    | O: 24h | O: 48h | O: 72h | O: 96h |
|          | E: 24h | E: 24h | E: 24h | E: 24h    | E: 24h | E: 24h | E: 24h | E: 24h |
| Nov. 10  | 66.6   | 49.1   | 49.0   | 38.7      | 0.677  | 0.713  | 0.716  | 0.716  |
| 11       | 75.2   | 80.6   | 78.4   | 78.4      | 0.410  | 0.621  | 0.623  | 0.623  |
| 12       | 75.1   | 45.2   | 56.5   | 55.8      | 0.373  | 0.406  | 0.425  | 0.430  |
| 13       | 72.8   | 58.0   | 52.9   | 63.5      | 0.298  | 0.534  | 0.581  | 0.589  |
| 14       | 75.6   | 54.6   | 53.8   | 53.1      | 0.285  | 0.567  | 0.593  | 0.609  |
| 15       | 82.7   | 43.9   | 42.4   | 44.6      | 0.105  | 0.554  | 0.589  | 0.571  |
| 16       | 69.8   | 34.3   | 28.4   | 27.4      | 0.146  | 0.464  | 0.507  | 0.505  |
| 17       |        |        |        |           |        |        |        |        |
|          | A: 96h | A: 96h | A: 96h | A: 96h    | A: 96h | A: 96h | A: 96h | A: 96h |
| Coverage | O: 24h | O: 48h | O: 72h | O: 96h    | O: 24h | O: 48h | O: 72h | O: 96h |
|          | E: 24h | E: 48h | E: 72h | E: 96h    | E: 24h | E: 48h | E: 72h | E: 96h |
| Nov. 10  | 53.0   | 62.1   | 62.1   | 60.7      | 0.746  | 0.657  | 0.657  | 0.577  |
| 11       | 54.7   | 61.9   | 63.7   | 08./      | 0.583  | 0.615  | 0.593  | 0.593  |
| 12       | 43.0   | 54.0   | 59.1   | 05.7      | 0.425  | 0.481  | 0.545  | 0.526  |
| 13       | 40.3   | 46.4   | 54.5   | 00.0      | 0.664  | 0.510  | 0.500  | 0.576  |
| 14       | 38.3   | 43.5   | 50.5   | 59.0      | 0.688  | 0.603  | 0.499  | 0.469  |
| 15       | 43.9   | 50.8   | 55.3   | 20.8      | 0.647  | 0.572  | 0.533  | 0.511  |
| 16       | 33.5   | 42.2   | 53.2   | 617 600   | 0.667  | 0.685  | 0.581  | 0.478  |
| 17       |        | 36.1   | 53.8   | 01./ 08.2 |        | 0.624  | 0.520  | 0.372  |
|          | A: 96h | A: 96h | A: 96h | A: 96h    | A: 96h | A: 96h | A: 96h | A: 96h |
| Coverage | O: 24h | O: 48h | O: 72h | O: 96h    | O: 24h | O: 48h | O: 72h | O: 96h |
| -        | E: 24h | E: 24h | E: 24h | E: 24h    | E: 24h | E: 24h | E: 24h | E: 24h |
| Nov. 10  | 53.6   | 47.1   | 47.1   | 38.7      | 0.746  | 0.715  | 0.715  | 0.716  |
| 11       | 58.1   | 81.0   | 78.4   | 78.4      | 0.668  | 0.624  | 0.623  | 0.623  |
| 12       | 41.3   | 48.0   | 56.7   | 55.8      | 0.489  | 0.432  | 0.433  | 0.430  |
| 13       | 39.5   | 49.8   | 55.4   | 63.5      | 0.691  | 0.612  | 0.588  | 0.589  |
| 14       | 39.5   | 45.8   | 51.9   | 53.1      | 0.698  | 0.612  | 0.612  | 0.609  |
| 15       | 37.8   | 43.1   | 43.1   | 44.6      | 0.682  | 0.552  | 0.566  | 0.571  |
| 16       | 33.7   | 29.6   | 27.9   | 27.4      | 0.614  | 0.542  | 0.510  | 0.505  |
| 17       |        |        |        |           |        |        |        |        |

Table R1 Sensitivity test for temporal coverage. Uses of analysis (A), observation (O), and evaluation (E) time windows (24 hours to 96 hours) were tested, and statistics, NRMSE and R, were compared. The best performance is marked as bold.

**2. Sensitivity tests for vertical layer selection**

For the sensitivity of vertical layer configuration, we added an extra test as the reviewer #2 suggested. With the maximum height fixed at 5,000m, we conducted a sensitivity test by increasing vertical layer resolution from 2 layers to 6 layers. As expected, we have better performance statistics as we increase the number of layers although the change is not dramatic after four layers (**Table R2**).

Indeed, this vertical layer sensitivity is one of the most important topics in the construction of smoke forecast system since it is directly related to the smoke plume rise problem. Therefore, we believe that the topic deserves a separate effort. We are preparing a follow-up study on the plume rise options, based on plume rise parameterization (in HYSPLIT) and inverse modeling technique. For current study, we like to narrow and focus on its own scope. While we demonstrate vertical layer selection sensitivity from its maximum height and vertical layer resolution, detailed analysis on the fire emission plume rise options warrants an additional full research.

|       | Date    | 2 layers
(100, 5000m) | 3 layers
(100, 2000,
5000m) | 4 layers
(100, 1000,
2000, 5000m) | 5 layers
(100, 1000,
1500, 2000,
5000m) | 6 layers
(100, 500, 1000,
1500, 2000,
5000m) |
|-------|---------|--------------------------|-----------------------------------|-----------------------------------------|--------------------------------------------------|-------------------------------------------------------|
| NRMSE | Nov. 10 | 72.9                     | 69.2                              | 65.9                                    | 63.7                                             | 63.7                                                  |
|       | 11      | 79.0                     | 71.6                              | 68.1                                    | 66.4                                             | 65.9                                                  |
|       | 12      | 70.7                     | 67.8                              | 65.5                                    | 65.0                                             | 63.5                                                  |
|       | 13      | 68.7                     | 67.6                              | 66.1                                    | 65.9                                             | 65.7                                                  |
|       | 14      | 72.9                     | 70.6                              | 67.2                                    | 67.0                                             | 66.3                                                  |
|       | 15      | 83.3                     | 77.1                              | 72.0                                    | 71.0                                             | 70.9                                                  |
|       | 16      | 78.9                     | 72.0                              | 73.2                                    | 72.5                                             | 72.5                                                  |
|       | 17      | 72.2                     | 70.6                              | 70.2                                    | 70.0                                             | 69.8                                                  |
| R     | Nov. 10 | 0.577                    | 0.593                             | 0.630                                   | 0.654                                            | 0.654                                                 |
|       | 11      | 0.490                    | 0.558                             | 0.598                                   | 0.614                                            | 0.620                                                 |
|       | 12      | 0.292                    | 0.285                             | 0.303                                   | 0.303                                            | 0.309                                                 |
|       | 13      | 0.449                    | 0.450                             | 0.468                                   | 0.470                                            | 0.476                                                 |
|       | 14      | 0.404                    | 0.419                             | 0.439                                   | 0.440                                            | 0.444                                                 |
|       | 15      | 0.252                    | 0.276                             | 0.338                                   | 0.346                                            | 0.348                                                 |
|       | 16      | 0.112                    | 0.164                             | 0.144                                   | 0.149                                            | 0.150                                                 |
|       | 17      | 0.198                    | 0.139                             | 0.139                                   | 0.139                                            | 0.146                                                 |

Table R2 Sensitivity tests for vertical layers resolution from two to six layers. The best performance is marked as bold.

The manuscript was revised including this information.

(Line 286) "Second, we tested the layers at which fire emissions are initiated in the model. As expected, including more layers results in better statistics, since the transport and dispersion of each smoke plume can vary with the altitude to which their fire emissions are allocated. We tested the model's uncertainties on layers' maximum extension and resolution, with varying selections of two to seven layers at 100, 500, 1000, 1500, 2000, 5000, or 10000 meters. To test the maximum extension, starting from two layers (i.e. with emissions released at 100 and 500 meters), we added the next higher layer over six test runs to investigate the effect of maximum extension of smoke plume. Figure 6b shows the results, and error statistics are summarized in Table 3. Including the 5000m layer, especially, resulted in noticeable changes, implying the potential benefit of including high-level transport for specific days. Since the 5000m layer is above typical planetary boundary layer height, emissions injected at this level experience different physical characteristics. Smoke lofted into the free troposphere is less affected by turbulence and scavenging, and transports easily hundreds or thousands of kilometers downwind because of the higher wind speeds. Addition of the 5000m layer would better represent the potential long-range transport. Smoke plume rise is one of traditionally important questions in smoke modelling, so further research on the topic is warranted. Effects of the layer resolution were also tested. Starting from two layers (i.e. 100m and 5000m), we added intermediate layers up to six layers, and evaluated their performances (Table S3). As expected, including more layers resulted in the better statistics, but its improvement was not significant after four layers. "

**3. Scope of the study**

Here, we like to address the scope of this manuscript again. In this study, we intended to conduct two goals: 1) Suggestion of inverse modeling framework, and 2) improvement of operational smoke forecasts. Our main priority here is to suggest a practical framework to help fire emission estimation. However, for the second goal, we do not claim that our new system is a complete solution and outperforms in all aspects because we know that many uncertainties remained to be resolved until we claim a clearly better forecast system. To improve the forecasting system, we admit that we need more improvement in other components, including the quality of satellite products (to monitor the dispersion of smoke plume), better meteorology and fire activity persistency, as we discussed in the manuscript.

We also provide point-by-point responses to each reviewer.

**REVIEWER #1**

- In section 1 (Introduction), there is no background information of applications of inverse modeling on fire emissions estimations. As this is the focus of this work, it would be helpful to inform readers about previous investigations on this topic, as well as to highlight the features and advantages of the inverse modeling system proposed in this work.

Thanks for the comment. We included some of previous inverse modeling efforts, but its application to fire emission estimation is limited. Most of fire emission estimations use a survey-based bottom-up or a satellite-based top-down approaches by simple comparison. While we agree that those efforts are valuable, we see that the inverse modeling technique is rarely used, especially for using smoke transport as a constraint. We included this information in the manuscript.

(Line 82) "Such an approach has been adapted to estimate inversely various emission sources, including greenhouse gas emissions (Kunik et al., 2019; Nickless et al., 2018; Turnbull et al., 2019), volcanic ashes and sulfur dioxide emissions (Boichu et al., 2014; Crawford et al., 2016; Zidikheri and Lucas, 2020), and radionuclide release from nuclear power plant incident (Chai et al., 2015; Katata et al., 2015; Li et al., 2019a), but was rarely used in fire emission estimation (e.g. Nikonovas et al., 2017)."

- The HYSPLIT model used to compute dispersion factors using the TCM approach is described in section 2.3, but some details are not provided, e.g., the temporal resolution of the HYSPLIT integration and the TCM results. Also, deposition considered by using a radioactive decay constant. Is it equivalent to the deposition process of fire smoke aerosols? How does it compare with the deposition considered in other Eulerian air quality models with more comprehensive chemistry?

Thanks for the comment. We included additional information of HYSPLIT configuration and output specification. For each day's inverse process, we saved HYSPLIT simulation outputs hourly, and ASDTA AOD were also produced hourly. The integration time step of HYSPLIT cannot be specified because it can vary during the simulation, being computed from the requirement that the advection distance per time-step should be less than the grid spacing. The maximum transport velocity is determined from the maximum transport speed during the previous hour. HYSPLIT handles dry and wet deposition of particles. Smoke particles are subject to dry deposition (including turbulent diffusion) in addition to gravitational settling and wet removal using meteorological model precipitation. Similar to other chemistry models, wet removal is defined as a scavenging coefficient to handle below- and within-cloud processes, although their parameterization would be model-specific. Please, see Stein et al. (2015) for detailed information of HYSPLIT procedures. Manuscript was revised to address the consistency to the previous system.

Radioactive decay is used for nuclear dispersion study. It is irrelevant for the study, so was removed.

(Line 130) "Hourly outputs were integrated to match with satellite observational data. HYSPLIT modelling options were configured to be consistent with the SFS system, including options for dry and wet depositions (i.e.  $0.8 \mu m$  diameter with 2 g/cm3 density) (Rolph et al., 2009).

- In the HYSPLIT simulations, dispersed concentrations were vertically integrated up to 5000 m to get partial column mass loading of smoke particles. Is there any reference for the selection of the column height (5000 m)? In addition, smoke loading from satellite observations is converted from AOD using a constant conversion factor. Could the authors further discuss the possible uncertainties that could be introduced by using this constant conversion factor? For example, how is this constant conversion factor compare with values reported by previous literature, and is there any relation between the conversion factor and other plume features (e.g., plume age)? And, this uncertainty can be considered in observation errors in the inverse modeling system.

Thanks for the comment. Most of HYSPLIT modeling settings in this study are configured to be consistent with the current operational system as described in Rolph et al. (2009).

With current design, the inverse system generates the best fit for given situation. Selection of conversion factor does affect the absolute values of both model and observation, but does not affect

their relative comparison (e.g. NRMSE, R). We have tested different conversion factors, and results are consistent in terms of their variations regardless of conversion factor assumption. However, it-should be noted that application of dynamic conversion factor (e.g. conversion factors varying depending on location and time) may affect the output, so better treatment of conversion factor will be required for more realistic inverse modeling process in the future. In current system, we include the observational error of AOD, but does not include error term from the conversion factor. We have included this information in the manuscript.

(L135) "Although we used a single conversion factor for the study, the actual conversion factors may vary in time and space (i.e.  $3.9 - 5.3 \text{ m}^2/\text{g}$ ) (Chand et al., 2006; Hobbs et al., 1996; Ichoku and Ellison, 2014; Nikonovas et al., 2017; Reid et al., 2005). Therefore, applying more realistic conversion factors and their uncertainties into the system would be another factor in the future improvement of the system."

- In section 3.2 of the cost function used in the inverse system, the estimations of error have not been provided, which are important terms for the inversion method. Firstly, how are the background error variances evaluated, and how does the value used here compare with typical uncertainties of fire emissions? Secondly, as mentioned in section 2, the observation error variances are composed of uncertainties in Lagrangian model, observations, as well as representative errors. It would be important to include more details about the determination of these error components. Thirdly, what kind of error terms should be considered in Fother and how are they determined?

Thanks for the comment. The manuscript has been modified to provide the estimations of error terms used in the cost function. The added text is shown below to address these issues. In addition, three references are added as well.

(Line 168) "A background term is included to measure the deviation of the emission estimate from its first guess,  $q_{ikt}^{b}$ , obtained from the operational BlueSky emission computation. The background term ensures the problem remains well-posed even with the limited observations available in certain circumstances. The background error variance  $\sigma_{ikt}^2$  measures uncertainties in  $q_{ikt}^b$ . Pan et al. (2020) compared six global emission estimates and found that the total emission differs by a factor of 3.8. However, emission estimations at specific locations and times can have much larger errors. In addition, the vertical distribution of the smoke emissions is difficult to determine and this adds even more uncertainties to the emission estimates. We chose a large uncertainty for the background term as  $\sigma_{ikt}$ =1000×qbikt+ 1000 kg/hr at all locations and heights to minimize the adverse impact of inaccurate BlueSky emission estimates. The observational error variances,  $\varepsilon_{nm}^2$ , represent uncertainties in both the model and observations, as well as the representative errors. Kondragunta et al. (2008) indicated that GOES aerosol retrievals over land were expected to have uncertainties within  $0.15\tau \pm 0.05$ , where  $\tau$  is the AOD. Paciorek et al. (2008) showed a better performance of GOES aerosol retrievals in eastern U.S. than in western U.S. Green et al. (2009) demonstrated that GOES AOD correlates best with AERONET in autumn (September to November) than in other seasons. They showed that the RMS error was 0.060 in autumn while the average for all seasons is 0.0149. Considering the better performance in the Eastern US and in November, AOD uncertainties of  $0.10\tau \pm 0.06$  are assumed in this paper. A slightly larger additive component of the AOD error is chosen to include the effects of the representative errors and model errors which do not vary with the observed AOD values.  $F_{other}$ refers to the other regularized terms that can be included in the cost function. For instance, Chai et al. (2015) has a temporal smoothness penalty term to avoid abrupt changes in the temporal profile of the release rates. While this optimization problem could be solved to obtain optimal emission estimates using many minimization tools, we used the Limited-Memory Broyden-Fletcher-Goldfarb-Shanno (BFGS) algorithm (Zhu et al., 1997)."

Christopher J. Paciorek, Yang Liu, Hortensia Moreno-Macias, and Shobha Kondragunta: Spatiotemporal Associations between GOES Aerosol Optical Depth Retrievals and Ground-Level PM2.5, Environmental Science & Technology 2008 42 (15), 5800-5806, DOI: 10.1021/es703181j Green M, Kondragunta S, Ciren P, Xu C. Comparison of GOES and MODIS aerosol optical depth (AOD) to aerosol robotic network (AERONET) AOD and IMPROVE PM2.5 mass at Bondville, Illinois. J Air Waste Manag Assoc. 2009;59(9):1082-1091. doi:10.3155/1047-3289.59.9.1082

Pan, X., Ichoku, C., Chin, M., Bian, H., Darmenov, A., Colarco, P., Ellison, L., Kucsera, T., da Silva, A., Wang, J., Oda, T., and Cui, G.: Six global biomass burning emission datasets: intercomparison and application in one global aerosol model, Atmos. Chem. Phys., 20, 969–994, https://doi.org/10.5194/acp-20-969-2020, 2020.

- The ASDTA smoke AOD data are assimilated in the HEIMS-fire system to obtain inverse estimation of fire emissions. As indicated in the introduction, the ASDTA data are based on automatic detections of fire smoke plume and represent smoke AOD, which means that the background AOD has been subtracted from total value. Is it correct? Then could the authors explain how is the background AOD derived? Also, the uncertainties in background AODs can impact inversion results, because an overestimation of background AOD will lead to underestimated smoke AOD (then underestimated fire emissions) and vice versa. This uncertainty can also be considered in the observation errors used for inversion.

Thanks for the comment. The ASDTA smoke AOD is not determined by subtracting the background AOD from the total AOD. The radiative signatures of an aerosol layer are determined by the scattering and absorption properties of the aerosol within a layer in the atmosphere. Reflectance value of the 0.86 µm channel (R0.86µm) divided by the reflectance value of the 0.63µm channel (R0.63µm) is often used to determine the aerosol type. Presence of smaller aerosols, like smoke, tends to reduce the values for this ratio, as smaller particles are more efficient at scattering light at 0.63µm. In addition, ASDTA also utilizes a pattern recognition technique to isolate smoke aerosols from other type of aerosols. The background AOD is not directly used to retrieve ASDTA smoke AOD. We utilize the observational error information of ASDTA AOD in the inverse modeling system.

(Line 111) "For each pixel, the radiative signatures of an aerosol layer (e.g. dust and smoke) are determined by the scattering and absorption properties of the aerosol. ASDTA also utilizes a pattern-recognition technique to isolate smoke aerosols from other type of aerosols, so it can recognize plumes transported far from fire sources."

- In section 3.3, the naming conventions of inversion and forecasting processes are described. It's a bit inaccurate to say that "fire emissions on November 13 can be estimated using ASDTA observations for 24 hours (i.e. oday=0), 48 hours (i.e. oday=-1), 72 hours (i.e. oday=-2), and 96 hours (i.e. oday=-3)." Technically, if we focus on fires on the target day (oday = 0), the regional impact and transported smoke plumes from those fires would be found on the following days (oday=1, 2, 3, etc.). Therefore, in this case the emissions on November 13 can only be constrained by the observations on oday=0 in reanalysis mode. On the other hand, the observations on oday=0 can constrain emissions on oday=0, as well as emissions on previous days simultaneously. So, the major benefit of including more observational days is getting more constraints for fires on multiple days, and providing a better estimation of the background smoke plume for the target day. In this case, it would be more precise to say that "for a target day of November 13, inversions are conducted using ASDTA observations for 24 hours (i.e. oday=0), 48 hours (i.e. oday=-1), 72 hours (i.e. oday=-2), and 96 hours (i.e. oday=-3)."

Thanks for the comment. Descriptions on the analytical and observational time windows are revised. Also, see discussions in the general response #1.

(Line 207) "This inverse system is designed to estimates fire emissions on the target day by analyzing past and present smoke field, and then utilizes them to forecast the future smoke field. The assimilation days (i.e. aday = 0,-1,-2) (see **Figure S1**) indicate the temporal coverage of dispersions and constraining observations. For a target day of November 13, inversions are conducted using HYSPLIT dispersion simulations and ASDTA observations for 24 hours (i.e. aday=0), 48 hours (i.e. aday=-1), 72 hours (i.e. aday=-2), and 96 hours (i.e. aday=-3). Estimated fire emissions are used to

**simulate fire smoke for November 13 (i.e. fday=0; reanalysis), and the same amount of fire emissions are used in forecast mode for November 14 (i.e. fday=+1) and 15 (i.e. fday=+2). "**

- Column particle mass loading is used to constrain fire smoke emissions, and emissions released at different numbers of vertical layers are tested in the sensitivity analysis. Including 5000 m level make an obvious improvement for the results. But there is a lack of analysis for the reason. A possible reason is that, 5000 m is usually above the Planetary Boundary Layer Height (PBLH). So, it would be interesting to examine the PBLH and plume injection height for this case, since smoke injection height is important to smoke transport. Smoke lofted into the free troposphere is often transported hundreds or thousands of kilometers downwind because of the higher wind speeds, generally lower turbulence levels, and less scavenging processes at higher altitudes. While, smoke trapped within the PBL is usually well mixed, and remains near the source region. If most of the fire spots in this case showed injected emissions above the PBLH, then it means that, including 5000 m in the simulations allows a better representation of emission injection, and the plume can be transported further and better constrained by observations.

**Thanks for the comment. We agree with the reviewer's comment. We included discussions on the vertical layer extension. Also, see the general response #2.**

(Line 292) "Including the 5000m layer, especially, resulted in noticeable changes, implying the potential benefit of including high-level transport for specific days. Since the 5000m layer is above typical planetary boundary layer height, emissions injected at this level experience different physical characteristics. Smoke lofted into the free troposphere is less affected by turbulence and scavenging, and transports easily hundreds or thousands of kilometers downwind because of the higher wind speeds. Addition of the 5000m layer would better represent the potential long-range transport."

**- As a follow-up of the last comment, could it provide better results by tuning the emission release heights incorporating information from a plume rise model?**

Thanks for the comment. We agree that plume rise models can help by providing 1) better initial guess of vertical allocation of emissions, and 2) complementary simulation when observations are limited. Please, also see the general response #2.

- For the sensitivity test on the time range of observation data used in inversion, "The 'one-day' (oday=0) simulation is run through the inverse model using dispersion and observations for the target day, while the 'two-day' simulation uses two days (i.e., 48 hours) of dispersion and observations (oday=-1)". As this sensitivity test focus on the time range of observations. I think it is unfair to compare the results using different days for both of dispersion and observations. For example, if we compare the results using 'one day' and 'two days' shown in the current test, for the 'one-day' simulation, all the mismatches of modeled and observed smoke mass loading would be attributed to adjustments of fires on the target day, which would likely lead to a significant error in the emission estimates for the target day. While the 'two-day' simulation allows the fires on oday=-1 to be constrained at the same time. Therefore, it would make more sense to use four days of dispersion for all the simulations for this test, and just change the observation days from 1 (24 hours) to 4 (96 hours).

**Thanks for the comment. We clarified the temporal coverage sensitivity test. Please, see the general response #2.**

- At the end of section 4.3, it is mentioned that "The November 17 output shows how the system responds when observations are limited or missing, although it still provides a robust result by honoring the initial guess information". But no result is referred. It would be clearer to add a reference to the figure/table supporting this sentence. If it is Figure 4, it would also be better to indicate the number of points in each panel to show that the observations are limited on Nov 17, given that many points could be overlapped.

Thanks for the comment. Although November 17 has no ASDTA AOD, the inverse system still produced a reasonable output (Figure 5). We included discussion on the performance of the inverse system with limited observational data.

(Line 25) "The November 17 output in **Figure 5** shows how the system responds when observations are limited or missing, although it still provides a robust result by honoring the initial guess information. On November 17, no ASDTA AOD was provided from the satellite operation. Under 48-hour configuration (i.e. aday=-1), the inverse system still produced reasonable outputs using limited observations (November 16) and initial guess emissions (November 16 and 17)."

- P8, L249-255, this paragraph can be more concise. Also, a description of the result of the spatial coverage sensitivity test is missing.

Thanks for the comment. We removed unnecessary sentences. We also included discussions on the spatial coverage sensitivity test result.

(Line 301) "In the third test, we varied the spatial coverage of input fire information. Although wildfire impacts easily spread by long-range transport, we could not include all the global fire information due to limited computational resources. We therefore tested different spatial domains of fire locations to evaluate what spatial coverage of wildfire detection information is required to estimate fire emissions. Fire sources inside domain 1 through 4 (Figure 3) were tested in the assimilation constrained by ASDTA AOD inside Domain 1. Figure 6c and Table 4 show correlation and error statistics from the sensitivity test of spatial coverage. In most days, we have better results when we include fire emission sources at least within domain 2. It makes sense considering the effects of transported fire plumes form Mississippi and Louisiana (Figure 3). Maximizing geographical coverage (e.g. domain 4) did not always result in the best performance in our case study. This result, however, should be taken carefully because we do not have strong fire activities outside domain 2 in our study case. Strong long-range transport cases, typically form northwestern US, Canada and Alaska, would have bigger impacts. "

- As stated in section 4.5, for the forecasting days, smoke is estimated as the summation of impact from fires existed on previous days and new emissions on the target days, and fire plumes initialized on fday =0, 1, 2 are used here for the summation. How do the authors determine how many previous days should be considered? Since the impact of fires can extend to multiple days, would it give a better result by adding the contribution of smokes initiated on the analysis days, e.g. oday = -1, -2, and -3?

Thanks for the comment. This is a key point of the temporal coverage sensitivity tests. Theoretically, we expect better output when we use longer temporal coverage, but it requires more computational resources. Current sensitivity test suggests that 48-hour time window can provide a decent result. In a forecast mode, we see the effect of persistency rate seems to be larger than other factors. Therefore, in the actual application of forecast system, more tuning of the system is required.

- In the abstract, it is concluded that the inverse modeling system developed here outperform than the current operational forecast product in terms of RMSE, but it's not clear RMSE of which variable is denoted here, and what observation dataset are the hindcast results and operational product evaluated against.

Thanks for the comment. Evaluations were conducted for smoke mass loading between hindcast and operational product against satellite-observed fire smoke mass loading for next 48-hour.

- P3, L92: "A modeling framework" -> "As a modeling framework"

**Corrected.**

- P4, L95: emission rate or emission? Do they represent the same term in this paper (i.e. fire smoke particles emission)? It seems that both are used throughout this paper. It would be better to use one of them and keep consistent.

**Thanks for the comment. Conceptually, it is 'emission' in general, and 'emission rate' is used as an actual input for the HYSPLIT. We used 'emission rate' specifically for cost function minimization or as a model input.**

- P4, L125: I do not quite understand why "the results shown in the study are obtained by multiplying the column height (i.e. 5000 m)" here. It has been shown that, the column TCM is calculated by integrating dispersed concentrations vertically, so the TCM is already in units of column loading per unit emission. Is it correct?

**Simulation outputs for TCM are in density unit and are converted to column mass loading by multiplying column height.**

- P7, L189: true color image. There is not a true color channel.

**Corrected.**

- P7, L214: "estimation of assimilated fire emissions" -> "estimation of fire emissions"

**Corrected.**

- Section 4.3, the time range of observations assimilated in this case experiment is not indicated.

**As mentioned in captions of Figure 4 & 5, 48-hour observations are used.**

- What's the date for the result shown in Figure 5?

**Dates are labeled in the left side of each panel. We specified the period in the caption.**

- P8, L225: remove "that"

**Revised.**

- P8, L242: "smoke dispersions" -> "fire emissions"

**Corrected.**

- P8, L247, "As expected, including more layers generally produce better result." This sentence is nearly a duplication of the sentence in L242-243.

**Thanks for the comment. We removed the sentence.**

- P10, L295: "From the top panel of Figure 9" -> "For the top panels of Figure 9"

**Corrected.**

- P10, L295: "are solely originated fires" -> "are solely originated from fires"

**Corrected.**

- P10, L303: "reply on" -> "rely on"

**Corrected.**

- P10, L312, for the "additional constraint", "transported smoke plume" could be better. There are other places of this term, please consider revising them accordingly.

**Thanks for the comment. We revised sentences.**

---

## Author Comment (AC2) · 28 Jun 2020

**"Inverse modeling of fire emissions constrained by smoke plume transport using HYSPLIT dispersion model and geostationary observations"**

**General response**

Authors express their appreciation to the two reviewers and the editor. Thanks to their productive comments, we were able to improve our manuscript. As the reviewer recommended, we have included additional tests and discussions on the sensitivity. Mistakes were revised, and performance statistics were also updated. We provide below the general responses and the point-by-point responses to the reviewer's comments. Reviewers' comments are shown in italics.

Here are three major points in the revised version of the manuscript.

1. Sensitivity tests for temporal coverage

As both reviewers commented, we agree that the temporal sensitivity section needs a clarification of the goal. Our original intention was to find the most efficient temporal coverage of the inverse system to conduct the assimilation process, which includes both time windows for the assimilation and the use of observational data. On the other hand, the reviewers asked to test the sensitivity by the change of observational data use. We believe that both tests are meaningful, so we included both test results in the manuscript. The former is important in terms of the efficiency of the inverse system because dispersion simulations to construct the TCM is the most time-consuming processes in the inverse system. The latter, selection of observations (used in the assimilation), is also important to investigate the responses of the inverse system.

Indeed, there are three adjustable time windows in the processes and evaluation of fire emission assimilation. As shown in Figure R1, the assimilation (or analysis) time window denotes the temporal coverage of fire emission sources and simulated dispersions. The observational data time window (or selection of observations used for constraining in the assimilation) can be set equal to or shorter than the assimilation time window. The evaluation time window (or selection of observations used for evaluation) is also set equal to or shorter than the observational time window. Therefore, we tested combinations of these time windows.

- 1) Assimilation time window (source, dispersion and assimilation process)
- 2) Observation time window (observation)
- 3) Evaluation time window (model and observation)

Model performance statistics were calculated with combinations of temporal coverages of assimilation (shown as "A" in Table R1; 24h-96h), observation (shown as "O", 24h-96h), and evaluation (shown as "E", 24h-96h). Results are summarized in Table R1. As discussed in the manuscript, including more analytical period generally improves the performance of the inverse system, but the difference is not critical for more than 48-hour time window.

With fixed analytical period (A:96h), performance statistics are better with shorter observational time window applied. It makes sense because we expect to have a better fit with smaller number of data points during the analysis process although it also has a risk of potential overfitting problem, as the reviewer #2 commented.

The manuscript was revised to include these additional sensitivity tests.

(Line 270) "First, we changed the assimilation time windows from one (24-hours) to four days (96-hours). Since the impact of fire emissions easily translates over multiple days, we tested how temporal coverage affects system results. The 'one-day' (aday=0) simulation is run through the inverse model using dispersions and observations for the target day, while the 'two-day' simulation uses two days (i.e., 48 hours) of dispersions and observations (aday=-1). For this test, all observations within the

assimilation time windows were selected for the assimilation and the evaluation. The results are shown in **Figure 6a**, while the correlation and error statistics are summarized in the top section of **Table 2** (i.e. [A:24h, O:24h, E:24h], ..., [A:96h, O:96h, E:96h]). With the exception of November 10 and 11, in the early stage of the fire event, both the correlation coefficient (R) and normalized root-mean-square error (NRMSE) were improved by the use of more days (i.e., three or four days) of dispersions and observations for the inverse model. This makes sense, because emissions from multi-day fire events spread out and affect concentrations over proceeding days.

A series of additional simulations were also conducted to test the system's sensitivity to the selection of observations for the assimilation and the evaluation. In these tests, we investigated combinations in assimilation time ("A" in **Table 2**), observational time ("O") and evaluation time windows ("E"). Results are also summarized in **Table 2**. For a fixed assimilation time period (i.e. [A:96h]), using shorter observational time window resulted in a better result. It is reasonable because we expect a better fitting with smaller number of data points. However, it can be easily exposed to overfitting problem if available data for the assimilation is too small."

Figure R1 Two steps for smoke forecast with the HEIMS-fire system. Temporal coverage of assimilation days (aday=0,-1, - 2,-3) and forecast days (fday=0,+1,+2) for operational tests. Assimilation process includes three time windows – assimilation, observation and evaluation time windows.

|          |        | NRMS   | SE (%) |           |        | I      | ર      |        |
|----------|--------|--------|--------|-----------|--------|--------|--------|--------|
|          | A: 24h | A: 48h | A: 72h | A: 96h    | A: 24h | A: 48h | A: 72h | A: 96h |
| Coverage | O: 24h | O: 48h | O: 72h | O: 96h    | O: 24h | O: 48h | O: 72h | O: 96h |
|          | E: 24h | E: 48h | E: 72h | E: 96h    | E: 24h | E: 48h | E: 72h | E: 96h |
| Nov. 10  | 66.6   | 63.7   | 62.7   | 68.7      | 0.677  | 0.654  | 0.669  | 0.577  |
| 11       | 75.2   | 65.9   | 64.0   | 63.7      | 0.410  | 0.620  | 0.587  | 0.593  |
| 12       | 75.1   | 63.5   | 61.4   | 60.6      | 0.373  | 0.309  | 0.562  | 0.526  |
| 13       | 72.8   | 65.7   | 59.9   | 59.6      | 0.298  | 0.476  | 0.397  | 0.576  |
| 14       | 75.6   | 66.3   | 61.0   | 56.8      | 0.285  | 0.444  | 0.522  | 0.469  |
| 15       | 82.7   | 70.9   | 62.7   | 60.2      | 0.105  | 0.348  | 0.465  | 0.511  |
| 16       | 69.8   | 72.5   | 68.2   | 61.7      | 0.146  | 0.150  | 0.372  | 0.478  |
| 17       |        | 69.8   | 72.5   | 68.2      |        | 0.146  | 0.150  | 0.372  |
|          | A: 24h | A: 48h | A: 72h | A: 96h    | A: 24h | A: 48h | A: 72h | A: 96h |
| Coverage | O: 24h | O: 48h | O: 72h | O: 96h    | O: 24h | O: 48h | O: 72h | O: 96h |
|          | E: 24h | E: 24h | E: 24h | E: 24h    | E: 24h | E: 24h | E: 24h | E: 24h |
| Nov. 10  | 66.6   | 49.1   | 49.0   | 38.7      | 0.677  | 0.713  | 0.716  | 0.716  |
| 11       | 75.2   | 80.6   | 78.4   | 78.4      | 0.410  | 0.621  | 0.623  | 0.623  |
| 12       | 75.1   | 45.2   | 56.5   | 55.8      | 0.373  | 0.406  | 0.425  | 0.430  |
| 13       | 72.8   | 58.0   | 52.9   | 63.5      | 0.298  | 0.534  | 0.581  | 0.589  |
| 14       | 75.6   | 54.6   | 53.8   | 53.1      | 0.285  | 0.567  | 0.593  | 0.609  |
| 15       | 82.7   | 43.9   | 42.4   | 44.6      | 0.105  | 0.554  | 0.589  | 0.571  |
| 16       | 69.8   | 34.3   | 28.4   | 27.4      | 0.146  | 0.464  | 0.507  | 0.505  |
| 17       |        |        |        |           |        |        |        |        |
|          | A: 96h | A: 96h | A: 96h | A: 96h    | A: 96h | A: 96h | A: 96h | A: 96h |
| Coverage | O: 24h | O: 48h | O: 72h | O: 96h    | O: 24h | O: 48h | O: 72h | O: 96h |
|          | E: 24h | E: 48h | E: 72h | E: 96h    | E: 24h | E: 48h | E: 72h | E: 96h |
| Nov. 10  | 53.0   | 62.1   | 62.1   | 60.7      | 0.746  | 0.657  | 0.657  | 0.577  |
| 11       | 54.7   | 61.9   | 63.7   | 08./      | 0.583  | 0.615  | 0.593  | 0.593  |
| 12       | 43.0   | 54.0   | 59.1   | 05.7      | 0.425  | 0.481  | 0.545  | 0.526  |
| 13       | 40.3   | 46.4   | 54.5   | 00.0      | 0.664  | 0.510  | 0.500  | 0.576  |
| 14       | 38.3   | 43.5   | 50.5   | 59.0      | 0.688  | 0.603  | 0.499  | 0.469  |
| 15       | 43.9   | 50.8   | 55.3   | 20.8      | 0.647  | 0.572  | 0.533  | 0.511  |
| 16       | 33.5   | 42.2   | 53.2   | 617 600   | 0.667  | 0.685  | 0.581  | 0.478  |
| 17       |        | 36.1   | 53.8   | 01./ 08.2 |        | 0.624  | 0.520  | 0.372  |
|          | A: 96h | A: 96h | A: 96h | A: 96h    | A: 96h | A: 96h | A: 96h | A: 96h |
| Coverage | O: 24h | O: 48h | O: 72h | O: 96h    | O: 24h | O: 48h | O: 72h | O: 96h |
| -        | E: 24h | E: 24h | E: 24h | E: 24h    | E: 24h | E: 24h | E: 24h | E: 24h |
| Nov. 10  | 53.6   | 47.1   | 47.1   | 38.7      | 0.746  | 0.715  | 0.715  | 0.716  |
| 11       | 58.1   | 81.0   | 78.4   | 78.4      | 0.668  | 0.624  | 0.623  | 0.623  |
| 12       | 41.3   | 48.0   | 56.7   | 55.8      | 0.489  | 0.432  | 0.433  | 0.430  |
| 13       | 39.5   | 49.8   | 55.4   | 63.5      | 0.691  | 0.612  | 0.588  | 0.589  |
| 14       | 39.5   | 45.8   | 51.9   | 53.1      | 0.698  | 0.612  | 0.612  | 0.609  |
| 15       | 37.8   | 43.1   | 43.1   | 44.6      | 0.682  | 0.552  | 0.566  | 0.571  |
| 16       | 33.7   | 29.6   | 27.9   | 27.4      | 0.614  | 0.542  | 0.510  | 0.505  |
| 17       |        |        |        |           |        |        |        |        |

Table R1 Sensitivity test for temporal coverage. Uses of analysis (A), observation (O), and evaluation (E) time windows (24 hours to 96 hours) were tested, and statistics, NRMSE and R, were compared. The best performance is marked as bold.

**2. Sensitivity tests for vertical layer selection**

For the sensitivity of vertical layer configuration, we added an extra test as the reviewer #2 suggested. With the maximum height fixed at 5,000m, we conducted a sensitivity test by increasing vertical layer resolution from 2 layers to 6 layers. As expected, we have better performance statistics as we increase the number of layers although the change is not dramatic after four layers (**Table R2**).

Indeed, this vertical layer sensitivity is one of the most important topics in the construction of smoke forecast system since it is directly related to the smoke plume rise problem. Therefore, we believe that the topic deserves a separate effort. We are preparing a follow-up study on the plume rise options, based on plume rise parameterization (in HYSPLIT) and inverse modeling technique. For current study, we like to narrow and focus on its own scope. While we demonstrate vertical layer selection sensitivity from its maximum height and vertical layer resolution, detailed analysis on the fire emission plume rise options warrants an additional full research.

|       | Date    | 2 layers
(100, 5000m) | 3 layers
(100, 2000,
5000m) | 4 layers
(100, 1000,
2000, 5000m) | 5 layers
(100, 1000,
1500, 2000,
5000m) | 6 layers
(100, 500, 1000,
1500, 2000,
5000m) |
|-------|---------|--------------------------|-----------------------------------|-----------------------------------------|--------------------------------------------------|-------------------------------------------------------|
| NRMSE | Nov. 10 | 72.9                     | 69.2                              | 65.9                                    | 63.7                                             | 63.7                                                  |
|       | 11      | 79.0                     | 71.6                              | 68.1                                    | 66.4                                             | 65.9                                                  |
|       | 12      | 70.7                     | 67.8                              | 65.5                                    | 65.0                                             | 63.5                                                  |
|       | 13      | 68.7                     | 67.6                              | 66.1                                    | 65.9                                             | 65.7                                                  |
|       | 14      | 72.9                     | 70.6                              | 67.2                                    | 67.0                                             | 66.3                                                  |
|       | 15      | 83.3                     | 77.1                              | 72.0                                    | 71.0                                             | 70.9                                                  |
|       | 16      | 78.9                     | 72.0                              | 73.2                                    | 72.5                                             | 72.5                                                  |
|       | 17      | 72.2                     | 70.6                              | 70.2                                    | 70.0                                             | 69.8                                                  |
| R     | Nov. 10 | 0.577                    | 0.593                             | 0.630                                   | 0.654                                            | 0.654                                                 |
|       | 11      | 0.490                    | 0.558                             | 0.598                                   | 0.614                                            | 0.620                                                 |
|       | 12      | 0.292                    | 0.285                             | 0.303                                   | 0.303                                            | 0.309                                                 |
|       | 13      | 0.449                    | 0.450                             | 0.468                                   | 0.470                                            | 0.476                                                 |
|       | 14      | 0.404                    | 0.419                             | 0.439                                   | 0.440                                            | 0.444                                                 |
|       | 15      | 0.252                    | 0.276                             | 0.338                                   | 0.346                                            | 0.348                                                 |
|       | 16      | 0.112                    | 0.164                             | 0.144                                   | 0.149                                            | 0.150                                                 |
|       | 17      | 0.198                    | 0.139                             | 0.139                                   | 0.139                                            | 0.146                                                 |

Table R2 Sensitivity tests for vertical layers resolution from two to six layers. The best performance is marked as bold.

The manuscript was revised including this information.

(Line 286) "Second, we tested the layers at which fire emissions are initiated in the model. As expected, including more layers results in better statistics, since the transport and dispersion of each smoke plume can vary with the altitude to which their fire emissions are allocated. We tested the model's uncertainties on layers' maximum extension and resolution, with varying selections of two to seven layers at 100, 500, 1000, 1500, 2000, 5000, or 10000 meters. To test the maximum extension, starting from two layers (i.e. with emissions released at 100 and 500 meters), we added the next higher layer over six test runs to investigate the effect of maximum extension of smoke plume. Figure 6b shows the results, and error statistics are summarized in Table 3. Including the 5000m layer, especially, resulted in noticeable changes, implying the potential benefit of including high-level transport for specific days. Since the 5000m layer is above typical planetary boundary layer height, emissions injected at this level experience different physical characteristics. Smoke lofted into the free troposphere is less affected by turbulence and scavenging, and transports easily hundreds or thousands of kilometers downwind because of the higher wind speeds. Addition of the 5000m layer would better represent the potential long-range transport. Smoke plume rise is one of traditionally important questions in smoke modelling, so further research on the topic is warranted. Effects of the layer resolution were also tested. Starting from two layers (i.e. 100m and 5000m), we added intermediate layers up to six layers, and evaluated their performances (Table S3). As expected, including more layers resulted in the better statistics, but its improvement was not significant after four layers. "

**3. Scope of the study**

Here, we like to address the scope of this manuscript again. In this study, we intended to conduct two goals: 1) Suggestion of inverse modeling framework, and 2) improvement of operational smoke forecasts. Our main priority here is to suggest a practical framework to help fire emission estimation. However, for the second goal, we do not claim that our new system is a complete solution and outperforms in all aspects because we know that many uncertainties remained to be resolved until we claim a clearly better forecast system. To improve the forecasting system, we admit that we need more improvement in other components, including the quality of satellite products (to monitor the dispersion of smoke plume), better meteorology and fire activity persistency, as we discussed in the manuscript.

We also provide point-by-point responses to each reviewer.

**REVIEWER #2**

1. How the experiments with different assimilation time windows are run needs to be made clearer. Is it the case that in all experiments the emissions are initiated at the same locations and times, the only difference being the satellite data assimilation time window? If that is the case, can you clarify why longer time windows results in better agreement with observations for the analysis? I would have thought it would be easier to fit the model to data from one timestep than to fit the model to data from different timesteps. I can not quite understand the explanation given in the paper (e.g. Line 240), which I interpret to mean that this is due to emissions at (x1,t1) affecting the smoke field at (x2,t2). This is true, but I would expect better results for the smoke field at (x2,t2) if there was no observation at (x1,t1) because the emission at (x1,t1) would only be constrained by the observation at (x2,t2) and not (x1,t1) and (x2,t2) simultaneously. You probably also need to clarify which data points are included in the verification statistics (for example, when you assimilate data on day 0, do you calculate verification statistics based on day 0 only or do you also include days -1, 2 etc?).

Thanks for the comment. We agree that the temporal coverage sensitivity section needs improvement. Please, see the general response #1.

2. In performing the forecasts, it is not completely clear whether you are using NWP analysis data to drive the forecasts, or you are using NWP forecast data. If this system is to be used operationally, then clearly NWP forecasts will need to be used, which we expect will result in poorer forecasts than when the NWP analyses are used. I think this needs to be clarified.

Thanks for the comment. This is a fair comment. At this point, we don't have an archive of full length of everyday NAM forecast, so our hindcast has an advantage over operational runs in terms of meteorology. We revised the manuscript to address it.

(Line 316) "Notable differences in the configuration of SFS and HEIMS are plume rise estimation, temporal resolution of fire emissions, fire decaying assumption, and meteorology. While SFS computes plume rise using the Briggs' equation (Arya, 1998; Briggs, 1969), which assumes an air parcel's rise is based only on the buoyance terms, HEIMS determines fire emissions' vertical allocation using an inverse system. At the initial guess, SFS fire emissions are evenly distributed in all layers. Current HEIMS assumes daily emission variation compared to hourly emissions of SFS. Also, SFS assumes 75% of emissions still happen at the same location the next day, the HEIMS uses 50% decay assumption after sensitivity tests, which will be discussed in the next section. For HEIMS simulation, we used aday=-1 (two-day temporal coverage) for the simulations shown. On the other hand, HEIMS would be benefitted with a better meteorology. Although both systems use the NAM12 forecast meteorology, HEIMS hindcast used the first 24 hours portion of everyday forecast cycle, and SFS used 72 hours forecast."

3. Nothing has been said about analysis and forecast errors. Can this system be used to predict errors (uncertainty) or is it purely deterministic? I think some comments on this issue in the paper would be useful.

Thanks for the comment. Uncertainties in the analysis processes include the quality of inputs (i.e. observations, dispersion, and meteorology). Although the system is not purely deterministic, it is difficult to estimate the uncertainties of the results without knowing the meteorological input errors. Performing an ensemble of HYSPLIT predictions using different meteorological inputs can provide some insight in this respect. We added the following text to the conclusion.

(Line 382) "It also should be noted that the uncertainties of the emission estimation and the smoke forecasts thereafter are not quantified in this study. An ensemble of HYSPLIT predictions using different meteorological inputs will be used to estimate the uncertainties of the results in the future."

1. In the abstract you have several acronyms that are only defined later in the text. These must be defined in the abstract or alternatively think about removing them altogether from the abstract. In my view they are not that important for the abstract and can be removed. For example, is it important to know that the model has been developed by NOAA in the abstract?

**Revised. We also changed "NOAA's HYSPLIT" to "HYSPLIT" (not an acronym, See Stein et al., 2015).**

2. Line 32: Remove capital letters from words 2-6 in the sentence starting with "Meeting...".

**Per convention, the acronym (NAAQS) has been added after "National Ambient Air Quality Standards".**

3. Line 43 (or thereabouts): You need to state that HYSPLIT is a Langrangian model (and maybe elaborate on what means) as in the next paragraph you start talking about Eularian models.

Thanks for the comment. We revised the sentence.

(Line 42) "The NOAA's HYSPLIT (Stein et al., 2015), a Lagrangian model which is designed to track air parcel trajectories, is then used to calculate transport, dispersion, and deposition of the emitted particulate matter. The SFS provides daily smoke forecasts over the continental United States, Alaska, and Hawaii to provide air-quality guidance to the public."

4. I also notice that you don't leave spaces between paragraphs, which makes it hard to read at times. Fix this if you can.

We are sorry for the inconvenience, but ACP template formats in such a way.

5. Line 54: You distinguish "top-down" and "bottom-up" approaches to estimating fire emissions. I think it would useful as well to discuss the advantages and disadvantages of these approaches somewhere in the introduction.

Thanks for the comment. We have added sentences for both approaches.

(Line 73) "Both bottom-up and top-down approaches have their own advantages and limitations. While the bottom-up approach may provide detailed information based on a process- or fuel-specific estimation, it relies on various surveys that require significant time and resources. On the other hand, the top-down approach relies on observations of a few atmospheric variables such as radiation or aerosol optical properties, but it has an advantage from its timely availability and geographical coverage. Both approaches complement each other for better fire emission estimation."

6. When discussing the methodology it might be worthwhile to explicitly mention that the "inverse system" enables the calculation of fire emissions that produces an "analysis" smoke field covering the observational time window (near past to present) and that these optimal emissions are then used to fore-cast the future smoke field. I do not think this has been made explicit although it may be obvious to the expert reader.

**Thanks for the comment. We have added sentences for two steps (emission inverse and forecast) for the system .**

**(L207) "This inverse system is designed to estimates fire emissions on the target day by analyzing past and present smoke field, and then utilizes them to forecast the future smoke field."**

7. Another point that I think needs to be made clear in the methodology section is how the experiments with different observational time windows (oday = 0, -1, 2 etc) differ from each other in the model setup. For example, for oday= 0, is it that there are no emissions prior to day 0, or is it that emissions prior to day0 are unconstrained (i.e. they use prior values), or is it that the emissions prior to day0 are in the model but they are constrained by observations on day 0 only. I suspect it is the last scenario, but this needs to be made clear. You could include this information in the paragraph starting at Line 177, which I think needs more clarity. For example, you could indicate that oday = 0 means that emissions from the model start time until day 0 (for example, November 13) are constrained by observations on day 0 and day -1 (November 12) etc. As I mentioned, it is not clear that is indeed the experimental setup, but whatever it is, it needs to made clear. When you calculate the verification statistics for the oday = 0 type experiment, for example, do you include all data (for days 0, -1, -2 etc) or just data for day 0? Please make this clearer.

**Thanks for the comment. Please, see the general response #1. We clarified the use of time windows for assimilation, observation and evaluation. Manuscript was also revised accordingly.**

8. It would also be useful in the context of Section 3 (or elsewhere like Figure 1 or in the conclusion) to say something about how the two systems (Blue Sky/Initial? and Inverse) really need to complement each other in practice. Apart from providing g the all-important first guess (or 'prior') to the inverse system, I would imagine that the Blue Sky system would also be needed to provide the first operational forecast when there are not yet any (or sufficient) observations to assimilate. In addition, in situations where there is extensive cloud cover, I would imagine that there would not be observations of smoke that could be used. This comment is related to Comment #5 above.

Thanks for the comment. This is true. Two systems complement each other. We included its importance at the manuscript.

(Line 263) "This case hints at the importance of both traditional (e.g. Blue Sky emissions) and new inverse system. They complement each other by one providing the latest data assimilation technique while the other providing a prior information and backup stability in a contaminated environment (e.g. excessive cloud cover)."

9. Line 219: The notation associated with this equation needs to be improved. Although I think I understand what you are trying to say, as it stands it is ambiguous. Firstly, it is not clear what "smoke" is supposed to be. I think you are looking at smoke mass loading, or it could be optical depth. Whatever it is, it needs to be made clear. I would use a single-letter symbol to represent the smoke variable. Since "c" is used in the first equation (Lines 148-149) that is what should be used here as well for consistency. Similarly, I would use a single letter to represent the TCM matrix coefficient (T seems like a good choice). Apart from that the equation doesn't look mathematically correct. Summing over the indices i, k, t should produce a scalar, independent of i, k, t but we know that the smoke field (mass load or optical depth) is a function of location, i, and time, t. So something is missing. I think some more work needs to be done here so that the equation makes sense.

**Thanks for the comment. We modified equations.**

(Line 248) "Using adjusted fire emissions, we can reconstruct the integrated smoke columns as a sum of adjusted emissions,  $q_{ikt}$ , applied to each TCM,  $T_{ikt}$ :

$$c(n,m) = \sum_{ikt} q_{ikt} \cdot T_{ikt}(n,m)$$

where i, k and t denote spatial, vertical and temporal allocation of emission sources, and m and n denote location and time of receptor (i.e. observation), respectively."

10. Line 226: change "during" to "when".

**Corrected.**

11. Line226: Last sentence of paragraph. Some more needs to be said here. Why is there no data on November 17? In addition, you are implying that no satellite data was used in the analysis (presumably because there was none – you must clearly state that if that is the case, and if so, why). And yet in the caption to Figure 5 we are told that a 48-hour observation time window was used. If that is the case, why could not the data for November 16 be used to constrain the analysis? Is it because your algorithm does not perform optimisation if any time steps are missing in the observational time window? You might have mentioned this elsewhere, but how frequent are the satellite observations? Every hour? Every 10 minutes?

**Thanks for the comment.**

ASDTA AODs are available for every hour, but some data are missing due to operational issues. We don't have control over it. In this simulation, we used 48-hour analysis. As the reviewer commented, we still have some data available to operate the inverse system. Therefore, this case demonstrated the case with limited available data. We revised manuscript to discuss the November 17 case.

(Line 259) "The November 17 output in **Figure 5** shows how the system responds when observations are limited or missing, although it still provides a robust result by honoring the initial guess information. On November 17, no ASDTA AOD was provided from the satellite operation. Under 48-hour configuration (i.e. aday=-1), the inverse system still produced reasonable outputs using limited observations (November 16) and initial guess emissions (November 16 and 17)."

12. Paragraph starting at Line 234: This is related to Comment #7. If you are using emissions initiated at the same time in all experiments in which you vary the length of observational time window, it is surprising, I think, that you get a better fit with a longer observational time window. Intuitively, all things being equal, you would expect to get a better fit with a smaller number of datapoints during the analysis. Of course, for the forecast you would expect better results with more data points during the analysis because you are less exposed to overfitting problems with more data points. It looks like you're talking about the analysis better on day 0 for example? I find that a bit unclear. I can see that if you are using data from all days to calculate your forecast verification statistics and not only on the days that data is assimilated, you would get better results by using more data because you are better constraining the smoke over a longer period. If that is the case, please make it clearer which data points are included in the verification statistics.

**Thanks for the comment. We strongly agree with the reviewer. In the revised manuscript, we included performance statistics with varying analysis, observation and evaluation time windows. Please, see the general response #1.**

13.Paragraph starting at Line 242: When I first read this, it sounded like you are varying the vertical resolution of the sources in these experiments. But is turns out you are just varying the maximum heights of the sources. I think you should consider rewording this so that it is immediately clear what you are doing. In some sense, I think varying the vertical resolution and keeping the maximum height the same would be a more interesting experiment. Rising plumes tend to have a neutral buoyancy level at which most of the mass tends to be detrained from the plume. This is true for volcanic plumes; I don't know if it's true here. So, changing the vertical resolution may have a significant impact on the results.

Thanks for the comment. We modified wording for the vertical layer sensitivity test.

(Line 290) "To test the maximum extension, starting from two layers (i.e. with emissions released at 100 and 500 meters), we added the next higher layer over six test runs to investigate the effect of maximum extension of smoke plume. **Figure 6b** shows the results, and correlation and error statistics are summarized in **Table 3.** Including the 5000m layer,"

We also conducted more sensitivity tests of vertical layers. Please, see the general response #2.

For the detrain level of smoke plume, some wildfire smoke plumes were reported to penetrate the PBL, but generally its upward motion is less dynamic compared to volcanic plumes. We believe the maximum level of smoke plume rise is still important, and added more sensitivity tests.

14. Line 255: You must say something about the outcome of the "third test" here. Would you say coverage does not matter? Or is it better for bigger coverage (Domains 3 and 4)? If the results are inconclusive, say so.

Thanks for the comment. We have added discussion on the spatial coverage test.

(Line 306) "In most days, we have better results when we include fire emission sources at least within domain 2. It makes sense considering the effects of transported fire plumes form Mississippi and Louisiana (**Figure 3**). Maximizing geographical coverage (e.g. domain 4) did not always result in the best performance in our case study. This result, however, should be taken carefully because we do not have strong fire activities outside domain 2 in our study case. Strong long-range transport cases, typically form northwestern US, Canada and Alaska, would have bigger impacts."

15. Line 271: You need to say something about "p" here even if briefly; for example, "p" is the persistence rate that measures...as explained further in Section xxx.

Thanks for the comment. We have added a description for p.

(Line 331) "The persistency rate, p, assumes the change of future day emissions. Its role will be discussed in the next Section."

16. Lines 298-307. Given the significant uncertainty in "p", an ensemble modelling approach might be fruitful in the forecast mode. Have you looked at this? If not, it might be something you could look at in the future.

Thanks for the comment. Currently, we are considering a dynamic modeling of decaying rate based on varying weather condition. Extending the idea to ensemble approach would be an excellent idea. We will further pursue the idea. We also included this idea in the manuscript.

(Line 381) "An ensemble of HYSPLIT predictions using different meteorological inputs will be used to estimate the uncertainties of the results in the future."

17. Line 325: What is "PM"?

PM denotes particulate matter (Line 25).